# Model Interpretability through the Lens of Computational Complexity

**Pablo Barceló**[1,4]**, Mikaël Monet**[2]**, Jorge Pérez**[3,4]**, Bernardo Subercaseaux**[3,4]

[1] Institute for Mathematical and Computational Engineering, PUC-Chile
[2] Inria Lille, France
[3] Department of Computer Science, Universidad de Chile
[4] Millennium Institute for Foundational Research on Data, Chile
pbarcelo@ing.puc.cl, mikael.monet@inria.fr, [jperez,bsuberca]@dcc.uchile.cl

## Abstract

In spite of several claims stating that some models are more interpretable than others – e.g., "linear models are more interpretable than deep neural networks" – we still lack a principled notion of interpretability to formally compare among different classes of models. We make a step towards such a notion by studying whether folklore interpretability claims have a correlate in terms of computational complexity theory. We focus on *local post-hoc explainability queries* that, intuitively, attempt to answer why individual inputs are classified in a certain way by a given model. In a nutshell, we say that a class $\mathcal{C}_1$ of models is *more interpretable* than another class $\mathcal{C}_2$, if the computational complexity of answering post-hoc queries for models in $\mathcal{C}_2$ is higher than for those in $\mathcal{C}_1$. We prove that this notion provides a good theoretical counterpart to current beliefs on the interpretability of models; in particular, we show that under our definition and assuming standard complexity-theoretical assumptions (such as $P \neq NP$), both linear and tree-based models are strictly more interpretable than neural networks. Our complexity analysis, however, does not provide a clear-cut difference between linear and tree-based models, as we obtain different results depending on the particular post-hoc explanations considered. Finally, by applying a finer complexity analysis based on parameterized complexity, we are able to prove a theoretical result suggesting that shallow neural networks are more interpretable than deeper ones.

## 1 Introduction

Assume a dystopian future in which the increasing number of submissions has forced journal editors to use machine-learning systems for automatically accepting or rejecting papers. Someone sends his/her work to the journal and the answer is a reject, so the person demands an explanation for the decision. The following are examples of three alternative ways in which the editor could provide an explanation for the rejection given by the system:

1. *In order to accept the submitted paper it would be enough to include a better motivation and to delete at least two mathematical formulas.*

2. *Regardless of the content and the other features of this paper, it was rejected because it has more than 10 pages and a font size of less than 11pt.*

3. *We only accept 1 out of 20 papers that do not cite any other paper from our own journal. In order to increase your chances next time, please add more references.*

These are examples of so called *local post-hoc explanations* [3, 19, 23, 26, 27]. Here, the term "local" refers to explaining the verdict of the system for a particular input [19, 27], and the term "post-hoc" refers to interpreting the system after it has been trained [23, 26]. Each one of the above explanations can be seen as a *query* asked about a system and an input for it. We call them *explainability queries*. The first query is related with the *minimum change required* to obtain a desired outcome ("what is the minimum change we must make to the article for it to be accepted by the system?"). The second one is known as a *sufficient reason* [32], and intuitively asks for a subset of the features of the given input that suffices to obtain the current verdict. The third one, that we call *counting completions*, relates to the probability of obtaining a particular output given the values in a subset of the features of the input.

In this paper we use explainability queries to formally compare the interpretability of machine-learning models. We do this by relating the interpretability of a class of models (e.g., decision trees) to the *computational complexity* of answering queries for models in that class. Intuitively the lower the complexity of such queries is, the more interpretable the class is. We study whether this intuition provides an appropriate correlate to folklore wisdom on the interpretability of models [20, 23, 28].

**Our contributions.** We formalize the framework described above (Section 2) and use it to perform a theoretical study of the computational complexity of three important types of explainability queries over three classes of models. We focus on models often mentioned in the literature as extreme points in the interpretability spectrum: decision trees, linear models, and deep neural networks. In particular, we consider the class of *free binary decision diagrams* (FBDDs), that generalize decision trees, the class of *perceptrons*, and the class of *multilayer perceptrons* (MLPs) with ReLU activation functions. The instantiation of our framework for these classes is presented in Section 3.

We show that, under standard complexity assumptions, the computational problems associated to our interpretability queries are strictly less complex for FBDDs than they are for MLPs. For instance, we show that for FBDDs, the queries minimum-change-required and counting-completions can be solved in polynomial time, while for MLPs these queries are, respectively, NP-complete and #P-complete (where #P is the prototypical intractable complexity class for counting problems). These results, together with results for other explainability queries, show that under our definition for comparing the interpretability of classes of models, FBDDs are indeed more interpretable than MLPs. This correlates with the folklore statement that tree-based models are more interpretable than deep neural networks. We prove similar results for perceptrons: most explainability queries that we consider are strictly less complex to answer for perceptrons than they are for MLPs. Since perceptrons are a realization of a linear model, our results give theoretical evidence for another folklore claim stating that linear models are more interpretable than deep neural networks. On the other hand, the comparison between perceptrons and FBDDs is not definitive and depends on the particular explainability query. We establish all our computational complexity results in Section 4.

Then, we observe that standard complexity classes are not enough to differentiate the interpretability of shallow and deep MLPs. To present a meaningful comparison, we then use the machinery of *parameterized complexity* [12, 16], a theory that allows the classification of hard computational problems on a finer scale. Using this theory, we are able to prove that there are explainability queries that are more difficult to solve for deeper MLPs compared to shallow ones, thus giving theoretical evidence that shallow MLPs are more interpretable. This is the most technically involved result of the paper, that we think provides new insights on the complexity of interpreting deep neural networks. We present the necessary concepts and assumptions as well as a precise statement of this result in Section 5.

Most definitions of interpretability in the literature are directly related to humans in a subjective manner [5, 10, 25]. In this respect we do not claim that our complexity-based notion of interpretability is *the* right notion of interpretability, and thus our results should be taken as a study of the correlation between a formal notion and the folklore wisdom regarding a subjective concept. We discuss this and other limitations of our results in Section 6. We only present a few sketches for proofs in the body of the paper and refer the reader to the appendix for detailed proofs of all our claims.

## 2    A framework to compare interpretability

In this section we explain the key abstract components of our framework. The idea is to introduce the necessary terminology to formalize our notion of being *more interpretable in terms of complexity*.

**Models and instances.** We consider an abstract definition of a model $\mathcal{M}$ simply as a Boolean function $\mathcal{M} : \{0,1\}^n \to \{0,1\}$. That is, we focus on binary classifiers with Boolean input features. Restricting inputs and outputs to be Booleans makes our setting cleaner while still covering several relevant practical scenarios. A class of models is just a way of grouping models together. An *instance* is a vector in $\{0,1\}^n$ and represents a possible input for a model. A *partial instance* is a vector in $\{0,1,\bot\}^n$, with $\bot$ intuitively representing "undefined" components. A partial instance $x \in \{0,1,\bot\}^n$ represents, in a compact way, the set of all instances in $\{0,1\}^n$ that can be obtained by replacing undefined components in $x$ with values in $\{0,1\}$. We call these the *completions* of $x$.

**Explainability queries.** An *explainability query* is a question that we ask about a model $\mathcal{M}$ and a (possibly partial) instance $x$, and refers to what the model $\mathcal{M}$ does on instance $x$. We assume all queries to be stated either as *decision problems* (that is, YES/NO queries) or as *counting problems* (queries that ask, for example, how many completions of a partial instance satisfy a given property). Thus, for now we can think of queries simply as functions having models and instances as inputs. We will formally define some specific queries in the next section, when we instantiate our framework.

**Complexity classes.** We assume some familiarity with the most common computational complexity classes of polynomial time (PTIME) and nondeterministic polynomial time (NP), and with the notion of hardness and completeness for complexity classes under polynomial time reductions. In the paper we also consider the class $\Sigma_2^p$, consisting of those problems that can be solved in NP if we further grant access to an oracle that solves NP queries in constant time. It is strongly believed that PTIME $\subsetneq$ NP $\subsetneq \Sigma_2^p$ [2], where for complexity classes $\mathcal{K}_1$ and $\mathcal{K}_2$ we have that $\mathcal{K}_1 \subsetneq \mathcal{K}_2$ means the following: problems in $\mathcal{K}_1$ can be solved in $\mathcal{K}_2$, but complete problems for $\mathcal{K}_2$ cannot be solved in $\mathcal{K}_1$.

While for studying the complexity of our decision problems the above classes suffice, for counting problems we will need another one. This will be the class #P, which corresponds to problems that can be defined as counting the number of accepting paths of a polynomial-time nondeterministic Turing machine [2]. Intuitively, #P is the counting class associated to NP: while the prototypical NP-complete problem is checking if a propositional formula is satisfiable (SAT), the prototypical #P-complete problem is counting how many truth assignments satisfy a propositional formula (#SAT). It is widely believed that #P is "harder" than $\Sigma_2^p$, which we write as $\Sigma_2^p \subsetneq$ #P.[1]

**Complexity-based interpretability of models.** Given an explainability query $Q$ and a class $\mathcal{C}$ of models, we denote by $Q(\mathcal{C})$ the computational problem defined by $Q$ restricted to models in $\mathcal{C}$. We define next the most important notion for our framework: that of being *more interpretable in terms of complexity* (*c-interpretable* for short). We will use this notion to compare among classes of models.

**Definition 1.** *Let $Q$ be an explainability query, and $\mathcal{C}_1$ and $\mathcal{C}_2$ be two classes of models. We say that $\mathcal{C}_1$ is strictly more c-interpretable than $\mathcal{C}_2$ with respect to $Q$, if the problem $Q(\mathcal{C}_1)$ is in the complexity class $\mathcal{K}_1$, the problem $Q(\mathcal{C}_2)$ is hard for complexity class $\mathcal{K}_2$, and $\mathcal{K}_1 \subsetneq \mathcal{K}_2$.*

For instance, in the above definition one could take $\mathcal{K}_1$ to be the PTIME class and $\mathcal{K}_2$ to be the NP class, or $\mathcal{K}_1 = $ NP and $\mathcal{K}_2 = \Sigma_2^p$.

## 3 Instantiating the framework and main results

Here we instantiate our framework on three important classes of Boolean models and explainability queries, and then present our main theorems comparing such models in terms of c-interpretability.

### 3.1 Specific models

**Binary decision diagrams.** A *binary decision diagram* (BDD [35]) is a rooted directed acyclic graph $\mathcal{M}$ with labels on edges and nodes, verifying: (i) each leaf is labeled with true or with false; (ii) each internal node (a node that is not a leaf) is labeled with an element of $\{1, \ldots, n\}$; and

(iii) each internal node has an outgoing edge labeled 1 and another one labeled 0. Every instance $\boldsymbol{x} = (x_1, \ldots, x_n) \in \{0, 1\}^n$ defines a unique path $\pi_{\boldsymbol{x}}$ from the root to a leaf in $\mathcal{M}$, which satisfies the following condition: for every non-leaf node $u$ in $\pi_{\boldsymbol{x}}$, if $i$ is the label of $u$, then the path $\pi_{\boldsymbol{x}}$ goes through the edge that is labeled with $x_i$. The instance $\boldsymbol{x}$ is positive, i.e., $\mathcal{M}(\boldsymbol{x}) := 1$, if the label of the leaf in the path $\pi_{\boldsymbol{x}}$ is true, and negative otherwise. The *size* $|\mathcal{M}|$ of $\mathcal{M}$ is its number of edges. A binary decision diagram $\mathcal{M}$ is *free* (FBDD) if for every path from the root to a leaf, no two nodes on that path have the same label. A *decision tree* is simply an FBDD whose underlying graph is a tree.

**Multilayer perceptron (MLP).** A multilayer perceptron $\mathcal{M}$ with $k$ layers is defined by a sequence of *weight* matrices $\boldsymbol{W}^{(1)}, \ldots, \boldsymbol{W}^{(k)}$, *bias* vectors $\boldsymbol{b}^{(1)}, \ldots, \boldsymbol{b}^{(k)}$, and *activation* functions $f^{(1)}, \ldots, f^{(k)}$. Given an instance $\boldsymbol{x}$, we inductively define

$$\boldsymbol{h}^{(i)} := f^{(i)}(\boldsymbol{h}^{(i-1)} \boldsymbol{W}^{(i)} + \boldsymbol{b}^{(i)}) \qquad (i \in \{1, \ldots, k\}), \tag{1}$$

assuming that $\boldsymbol{h}^{(0)} := \boldsymbol{x}$. The output of $\mathcal{M}$ on $\boldsymbol{x}$ is defined as $\mathcal{M}(\boldsymbol{x}) := \boldsymbol{h}^{(k)}$. In this paper we assume all weights and biases to be rational numbers. That is, we assume that there exists a sequence of positive integers $d_0, d_1, \ldots, d_k$ such that $\boldsymbol{W}^{(i)} \in \mathbb{Q}^{d_{i-1} \times d_i}$ and $\boldsymbol{b}^{(i)} \in \mathbb{Q}^{d_i}$. The integer $d_0$ is called the *input size* of $\mathcal{M}$, and $d_k$ the *output size*. Given that we are interested in binary classifiers, we assume that $d_k = 1$. We say that an MLP as defined above has $(k-1)$ *hidden layers*. The *size* of an MLP $\mathcal{M}$, denoted by $|\mathcal{M}|$, is the total size of its weights and biases, in which the size of a rational number $p/q$ is $\log_2(p) + \log_2(q)$ (with the convention that $\log_2(0) = 1$).

We focus on MLPs in which all internal functions $f^{(1)}, \ldots, f^{(k-1)}$ are the ReLU function $\mathrm{relu}(x) := \max(0, x)$. Usually, MLP binary classifiers are trained using the *sigmoid* as the output function $f^{(k)}$. Nevertheless, when an MLP classifies an input (after training), it takes decisions by simply using the *pre activations*, also called *logits*. Based on this and on the fact that we only consider already trained MLPs, we can assume without loss of generality that the output function $f^{(k)}$ is the *binary step* function, defined as $\mathrm{step}(x) := 0$ if $x < 0$, and $\mathrm{step}(x) := 1$ if $x \geq 0$.

**Perceptron.** A perceptron is an MLP with no hidden layers (i.e., $k = 1$). That is, a perceptron $\mathcal{M}$ is defined by a pair $(\boldsymbol{W}, \boldsymbol{b})$ such that $\boldsymbol{W} \in \mathbb{Q}^{d \times 1}$ and $\boldsymbol{b} \in \mathbb{Q}$, and the output is $\mathcal{M}(\boldsymbol{x}) = \mathrm{step}(\boldsymbol{x}\boldsymbol{W} + \boldsymbol{b})$. Because of its particular structure, a perceptron is usually defined as a pair $(\boldsymbol{w}, b)$ with $\boldsymbol{w}$ a rational vector and $b$ a rational number. The output of $\mathcal{M}(\boldsymbol{x})$ is then 1 if and only if $\langle \boldsymbol{x}, \boldsymbol{w} \rangle + b \geq 0$, where $\langle \boldsymbol{x}, \boldsymbol{w} \rangle$ denotes the dot product between $\boldsymbol{x}$ and $\boldsymbol{w}$.

## 3.2 Specific queries

Given instances $\boldsymbol{x}$ and $\boldsymbol{y}$, we define $\mathrm{d}(\boldsymbol{x}, \boldsymbol{y}) := \sum_{i=1}^n |\boldsymbol{x}_i - \boldsymbol{y}_i|$ as the number of components in which $\boldsymbol{x}$ and $\boldsymbol{y}$ differ. We now formalize the minimum-change-required problem, which checks if the output of the model can be changed by flipping the value of at most $k$ components in the input.

| | |
|---|---|
| Problem: | MINIMUMCHANGEREQUIRED (MCR) |
| Input: | Model $\mathcal{M}$, instance $\boldsymbol{x}$, and $k \in \mathbb{N}$ |
| Output: | YES, if there exists an instance $\boldsymbol{y}$ with $\mathrm{d}(\boldsymbol{x}, \boldsymbol{y}) \leq k$ and $\mathcal{M}(\boldsymbol{x}) \neq \mathcal{M}(\boldsymbol{y})$, and NO otherwise |

Notice that, in the above definition, instead of "finding" the minimum change we state the problem as a YES/NO query (a decision problem) by adding an additional input $k \in \mathbb{N}$ and then asking for a change of size at most $k$. This is a standard way of stating a problem to analyze its complexity [2]. Moreover, in our results, when we are able to solve the problem in PTIME then we can also output a minimum change, and it is clear that if the decision problem is hard then the optimization problem is also hard. Hence, we can indeed state our problems as decision problems without loss of generality.

To introduce our next query, recall that a partial instance is a vector $\boldsymbol{y} = (y_1, \ldots, y_n) \in \{0, 1, \bot\}^n$, and a completion of it is an instance $\boldsymbol{x} = (x_1, \ldots, x_n) \in \{0, 1\}^n$ such that for every $i$ where $y_i \in \{0, 1\}$ it holds that $x_i = y_i$. That is, $\boldsymbol{x}$ coincides with $\boldsymbol{y}$ on all the components of $\boldsymbol{y}$ that are not $\bot$. Given an instance $\boldsymbol{x}$ and a model $\mathcal{M}$, a *sufficient reason for $\boldsymbol{x}$ with respect to $\mathcal{M}$* [32] is a partial instance $\boldsymbol{y}$, such that $\boldsymbol{x}$ is a completion of $\boldsymbol{y}$ and every possible completion $\boldsymbol{x}'$ of $\boldsymbol{y}$ satisfies $\mathcal{M}(\boldsymbol{x}') = \mathcal{M}(\boldsymbol{x})$. That is, knowing the value of the components that are defined in $\boldsymbol{y}$ is

enough to determine the output $\mathcal{M}(\boldsymbol{x})$. Observe that an instance $\boldsymbol{x}$ is always a sufficient reason for itself, and that $\boldsymbol{x}$ could have multiple (other) sufficient reasons. However, given an instance $\boldsymbol{x}$, the sufficient reasons of $\boldsymbol{x}$ that are most interesting are those having the least possible number of defined components; indeed, it is clear that the less defined components a sufficient reason has, the more information it provides about the decision of $\mathcal{M}$ on $\boldsymbol{x}$. For a partial instance $\boldsymbol{y}$, let us write $\|\boldsymbol{y}\|$ for its number of components that are not $\bot$. The previous observations then motivate our next interpretability query.

---

Problem: MINIMUMSUFFICIENTREASON (MSR)
Input: Model $\mathcal{M}$, instance $\boldsymbol{x}$, and $k \in \mathbb{N}$
Output: YES, if there exists a sufficient reason $\boldsymbol{y}$ for $\boldsymbol{x}$ wrt. $\mathcal{M}$ with $\|\boldsymbol{y}\| \leq k$, and NO otherwise

---

As for the case of MCR, notice that we have formalized this interpretability query as a decision problem. The last query that we will consider refers to counting the number of positive completions for a given partial instance.

---

Problem: COUNTCOMPLETIONS (CC)
Input: Model $\mathcal{M}$, partial instance $\boldsymbol{y}$
Output: The number of completions $\boldsymbol{x}$ of $\boldsymbol{y}$ such that $\mathcal{M}(\boldsymbol{x}) = 1$

---

Intuitively, this query informs us on the proportion of inputs that are accepted by the model, given that some particular features have been fixed; or, equivalently, on the *probability* that such an instance is accepted, assuming the other features to be uniformly and independently distributed.

## 3.3 Main interpretability theorems

We can now state our main theorems, which are illustrated in Figure 1. In all these theorems we use $\mathcal{C}_{\text{MLP}}$ to denote the class of all models (functions from $\{0,1\}^n$ to $\{0,1\}$) that are defined by MLPs, and similarly for $\mathcal{C}_{\text{FBDD}}$ and $\mathcal{C}_{\text{Perceptron}}$. The proofs for all these results will follow as corollaries from the detailed complexity analysis that we present in Section 4. We start by stating a strong separation between FBDDs and MLPs, which holds for all the queries presented above.

**Theorem 2.** $\mathcal{C}_{\text{FBDD}}$ *is strictly more c-interpretable than* $\mathcal{C}_{\text{MLP}}$ *with respect to* MCR*,* MSR*, and* CC*.*

For the comparison between perceptrons and MLPs, we can establish a strict separation for MCR and MSR , but not for CC. In fact, CC has the same complexity for both classes of models, which means that none of these classes strictly "dominates" the other in terms of c-interpretability for CC.

**Theorem 3.** $\mathcal{C}_{\text{Perceptron}}$ *is strictly more c-interpretable than* $\mathcal{C}_{\text{MLP}}$ *with respect to* MCR *and* MSR*. In turn, the problems* $\text{CC}(\mathcal{C}_{\text{Perceptron}})$ *and* $\text{CC}(\mathcal{C}_{\text{MLP}})$ *are both complete for the same complexity class.*

The next result shows that, in terms of c-interpretability, the relationship between FBDDs and perceptrons is not clear, as each one of them is strictly more c-interpretable than the other for some explainability query.

**Theorem 4.** *The problems* $\text{MCR}(\mathcal{C}_{\text{FBDD}})$ *and* $\text{MCR}(\mathcal{C}_{\text{Perceptrons}})$ *are both in* PTIME*. However,* $\mathcal{C}_{\text{Perceptron}}$ *is strictly more c-interpretable than* $\mathcal{C}_{\text{FBDD}}$ *with respect to* MSR*, while* $\mathcal{C}_{\text{FBDD}}$ *is strictly more c-interpretable than* $\mathcal{C}_{\text{Perceptron}}$ *with respect to* CC*.*

We prove these results in the next section, where for each query $Q$ and class of models $\mathcal{C}$ we pinpoint the exact complexity of the problem $Q(\mathcal{C})$.

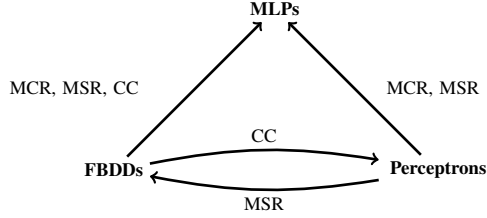

Figure 1: Illustration of the main interpretability results. Arrows depict that the pointed class of models is harder with respect to the query that labels the edge. We omit labels (or arrows) when a problem is complete for the same complexity class for two classes of models.

# 4    The complexity of explainability queries

|  | FBDDs | Perceptrons | MLPs |
|---|---|---|---|
| MINIMUMCHANGEREQUIRED | PTIME | PTIME | NP-complete |
| MINIMUMSUFFICIENTREASON | NP-complete | PTIME | $\Sigma_2^p$-complete |
| CHECKSUFFICIENTREASON | PTIME | PTIME | coNP-complete |
| COUNTCOMPLETIONS | PTIME | #P-complete | #P-complete |

Table 1: Summary of our complexity results.

In this section we present our technical complexity results proving Theorems 2, 3, and 4. We divide our results in terms of the queries that we consider. We also present a few other complexity results that we find interesting on their own. A summary of the results is shown in Table 1. With the exception of Proposition 6, items (1) and (3), the proofs for this section are relatively routine, were already known or follow from known techniques. As mentioned in the introduction, we only present the main ideas of some of the proofs in the body of the paper, and a detailed exposition of each result can be found in the appendix.

## 4.1    The complexity of MINIMUMCHANGEREQUIRED

In what follows we determine the complexity of the MINIMUMCHANGEREQUIRED problem for the three classes of models that we consider.

**Proposition 5.** *The* MINIMUMCHANGEREQUIRED *query is (1) in* PTIME *for FBDDs, (2) in* PTIME *for perceptrons, and (3)* NP*-complete for MLPs.*

*Proof sketch.* This query has been shown to be solvable in PTIME for *ordered* binary decision diagrams (OBDDs, a restricted form of FBDDs) by Shih et al. [31, Theorem 6] (the query is called *robustness* in the work of Shih et al. [31]). We show that the same proof applies to FBDDs. Recall that in an FBDD every internal node is labeled with a feature index in $\{1, \ldots, n\}$. The main idea is to compute a quantity $\mathrm{mcr}_u(\boldsymbol{x}) \in \mathbb{N} \cup \{\infty\}$ for every node $u$ of the FBDD $\mathcal{M}$. This quantity represents the minimum number of features that we need to flip in $\boldsymbol{x}$ to modify the classification $\mathcal{M}(\boldsymbol{x})$ if we are only allowed to change features associated with the paths from $u$ to some leaf in the FBDD. One can easily compute these values by processing the FBDD bottom-up. Then the minimum change required for $\boldsymbol{x}$ is the value $\mathrm{mcr}_r(\boldsymbol{x})$ where $r$ is the root of $\mathcal{M}$, and thus we simply return YES if $\mathrm{mcr}_r(\boldsymbol{x}) \leq k$, and NO otherwise.

For the case of a perceptron $\mathcal{M} = (\boldsymbol{w}, b)$ and of an instance $\boldsymbol{x}$, let us assume without loss of generality that $\mathcal{M}(\boldsymbol{x}) = 1$. We first define the *importance* $s(i) \in \mathbb{Q}$ of every input feature at position $i$ as follows: if $x_i = 1$ then $s(i) := w_i$, and if $x_i = 0$ then $s(i) := -w_i$. Consider now the set $S$ that contains the top $k$ most important input features for which $s(i) > 0$. We can easily show that it is enough to check whether flipping every feature in $S$ changes the classification of $\boldsymbol{x}$, in which case we return YES, and return NO otherwise.

Finally, NP membership of MCR for MLPs is clear: guess a partial instance $\boldsymbol{y}$ with $\mathrm{d}(\boldsymbol{x}, \boldsymbol{y}) \leq k$ and check in polynomial time that $\mathcal{M}(\boldsymbol{x}) \neq \mathcal{M}(\boldsymbol{y})$. We prove hardness with a simple reduction from the VERTEXCOVER problem for graphs, which is known to be NP-complete. □

Notice that this result immediately yields Theorems 2, 3, and 4 for the case of MCR.

## 4.2 The complexity of MINIMUMSUFFICIENTREASON

We now study the complexity of MINIMUMSUFFICIENTREASON. The following result yields Theorems 2, 3, and 4 for the case of MSR.

**Proposition 6.** *The* MINIMUMSUFFICIENTREASON *query is (1)* NP-*complete for FBDDs (and hardness holds already for decision trees), (2) in* PTIME *for perceptrons, and (3)* $\Sigma_2^p$-*complete for MLPs.*

*Proof sketch.* Membership of the problem in the respective classes is easy. We show NP-completeness of the problem for FBDDs by a nontrivial reduction from the NP-complete problem of determining whether a directed acyclic graph has a dominating set of size at most $k$ [22]. For a perceptron $\mathcal{M} = (\boldsymbol{w}, b)$ and an instance $\boldsymbol{x}$, assume without loss of generality that $\mathcal{M}(\boldsymbol{x}) = 1$. As in the proof of Proposition 5, we consider the importance of every component of $\boldsymbol{x}$, and prove that it is enough to check whether the $k$ most important features of $\boldsymbol{x}$ are a sufficient reason for it, in which case we return YES, and simply return NO otherwise. Finally, the $\Sigma_2^p$-completeness for MLPs is obtained again using a technical reduction from the problem called SHORTEST IMPLICANT CORE, defined and shown to be $\Sigma_2^p$-complete by Umans [34]. □

To refine our analysis, we also consider the natural problem of *checking* if a given partial instance is a sufficient reason for an instance.

| | |
|---|---|
| Problem: | CHECKSUFFICIENTREASON (CSR) |
| Input: | Model $\mathcal{M}$, instance $\boldsymbol{x}$ and a partial instance $\boldsymbol{y}$ |
| Output: | YES, if $\boldsymbol{y}$ is a sufficient reason for $\boldsymbol{x}$ wrt. $\mathcal{M}$, and NO otherwise |

We obtain the following (easy) result.

**Proposition 7.** *The query* CHECKSUFFICIENTREASON *is (1) in* PTIME *for FBDDs, (2) in* PTIME *for perceptrons, and (3)* co-NP-*complete for MLPs.*

We note that this result for FBDDs already appears in [9] (under the name of *implicant check*). Interestingly, we observe that this new query maintains the comparisons in terms of c-interpretability, in the sense that $\mathcal{C}_{\text{FBDD}}$ and $\mathcal{C}_{\text{Perceptron}}$ are strictly more c-interpretable than $\mathcal{C}_{\text{MLP}}$ with respect to CSR.

## 4.3 The complexity of COUNTCOMPLETIONS

What follows is our main complexity result regarding the query COUNTCOMPLETIONS, which yields Theorems 2, 3, and 4 for the case of CC.

**Proposition 8.** *The query* COUNTCOMPLETIONS *is (1) in* PTIME *for FBDDs, (2)* #P-*complete for perceptrons, and (3)* #P-*complete for MLPs.*

*Proof sketch.* Claim (1) is a a well-known fact that is a direct consequence of the definition of FBDDs; indeed, we can easily compute by bottom-up induction of the FBDD a quantity representing for each node the number of positive completions of the sub-FBDD rooted at that node (e.g., see [9, 35]). We prove (2) by showing a reduction from the #P-complete problem #KNAPSACK, i.e., counting the number of solutions to a $0/1$ knapsack input.[2] For the last claim, we show that MLPs with ReLU activations can simulate arbitrary Boolean formulas, which allows us to directly conclude (3) since counting the number of satisfying assignments of a Boolean formula is #P-complete. □

**Comparing perceptrons and MLPs.** Although the query COUNTCOMPLETIONS is #P-complete for perceptrons, we can still show that the complexity goes down to PTIME if we assume the weights and biases to be integers given in unary; this is commonly called *pseudo-polynomial time*.

**Proposition 9.** *The query* COUNTCOMPLETIONS *can be solved in pseudo-polynomial time for perceptrons (assuming the weights and biases to be integers given in unary).*

*Proof sketch.* This is proved by first reducing the problem to #KNAPSACK, and then using a classical dynamic programming algorithm to solve #KNAPSACK in pseudo-polynomial time. □

This result establishes a difference between perceptrons and MLPs in terms of CC, as this query remains #P-complete for the latter even if weights and biases are given as integers in unary. Another difference is established by the fact that COUNTCOMPLETIONS for perceptrons can be efficiently approximated, while this is not the case for MLPs. To present this idea, we briefly recall the notion of *fully polynomial randomized approximation scheme* (FPRAS [21]), which is heavily used to refine the analysis of the complexity of #P-hard problems. Intuitively, an FPRAS is a polynomial time algorithm that computes with high probability a $(1 - \epsilon)$-multiplicative approximation of the exact solution, for $\epsilon > 0$, in polynomial time in the size of the input and in the parameter $1/\epsilon$. We show:

**Proposition 10.** *The problem* COUNTCOMPLETIONS *restricted to perceptrons admits an FPRAS (and the use of randomness is not even needed in this case). This is not the case for MLPs, on the other hand, at least under standard complexity assumptions.*

## 5 Parameterized results for MLPs in terms of number of layers

In Section 4.1 we proved that the query MINIMUMCHANGEREQUIRED is NP-complete for MLPs. Moreover, a careful inspection of the proof reveals that MCR is already NP-hard for MLPs with only a few layers. This is not something specific to MCR: in fact, all lower bounds for the queries studied in the paper in terms of MLPs hold for a small, fixed number of layers. Hence, we cannot differentiate the interpretability of shallow and deep MLPs with the complexity classes that we have used so far.

In this section, we show how to construct a gap between the (complexity-based) interpretability of shallow and deep MLPs by considering refined complexity classes in our $c$-interpretability framework. In particular, we use *parameterized complexity* [12, 16], a branch of complexity theory that studies the difficulty of a problem in terms of multiple input parameters. To the best of our knowledge, the idea of using parameterized complexity theory to establish a gap in the complexity of interpreting shallow and deep networks is new.

We first introduce the main underlying idea of parameterized complexity in terms of two classical graph problems: VERTEXCOVER and CLIQUE. In both problems the input is a pair $(G, k)$ with $G$ a graph and $k$ an integer. In VERTEXCOVER we verify if there exists a set of nodes of size at most $k$ that includes at least one endpoint for every edge in $G$. In CLIQUE we check if there exists a set of nodes of size at most $k$ such that all nodes in the set are adjacent to each other. Both problems are known to be NP-complete. However, this analysis treats $G$ and $k$ at the same level, which might not be fair in some practical situations in which $k$ is much smaller than the size of $G$. Parameterized complexity then studies how the complexity of the problems behaves when the input is only $G$, and $k$ is regarded as a small *parameter*.

It happens to be the case that VERTEXCOVER and CLIQUE, while both NP-complete, have a different status in terms of parameterized complexity. Indeed, VERTEXCOVER can be solved in time $O(2^k \cdot |G|)$, which is polynomial in the size of the input $G$ – with the exponent not depending on $k$ – and, thus, it is called *fixed-parameter tractable* [12]. In turn, it is widely believed that there is no algorithm for CLIQUE with time complexity $O(f(k) \cdot \text{poly}(G))$ – with $f$ being any computable function, that depends only on $k$ – and thus it is *fixed-parameter intractable* [12]. To study the notion of fixed-parameter intractability, researchers on parameterized complexity have introduced the W[$t$] complexity classes (with $t \geq 1$), which form the so called W-*hierarchy*. For instance CLIQUE is W[1]-complete [12]. A core assumption in parameterized complexity is that W[$t$] $\subsetneq$ W[$t + 1$], for every $t \geq 1$.

In this paper we will use a related hierarchy, called the W(Maj)-hierarchy [14]. We defer the formal definitions of these two hierachies to the appendix. We simply mention here that both classes, W[$t$] and W(Maj)[$t$], are closely related to logical circuits of depth $t$. The circuits that define the W-hierarchy use gates AND, OR and NOT, while circuits for W(Maj) use only the MAJORITY gate (which outputs a 1 if more than half of its inputs are 1). Our result below applies to a special class of MLPs that we call restricted-MLPs (rMLPs for short), where we assume that the number of digits of each weight and bias in the MLP is at most logarithmic in the number of neurons in the MLP (a detailed exposition of this restriction can be found in the appendix). We can now formally state the main result of this section.

**Proposition 11.** *For every $t \geq 1$ the* MINIMUMCHANGEREQUIRED *query over rMLPs with $3t + 3$ layers is* $W(\mathsf{Maj})[t]$-*hard and is contained in* $W(\mathsf{Maj})[3t + 7]$.

By assuming that the $W(\mathsf{Maj})$-hierarchy is strict, we can use Proposition 11 to provide separations for rMLPs with different numbers of layers. For instance, instantiating the above result with $t = 1$ we obtain that for rMLPs with 6 layers, the MCR problem is in $W(\mathsf{Maj})[3t + 7] = W(\mathsf{Maj})[10]$. Moreover, instantiating it with $t = 11$ we obtain that for rMLPs with 36 layers, the MCR problem is $W(\mathsf{Maj})[11]$-hard. Thus, assuming that $W(\mathsf{Maj})[10] \subsetneq W(\mathsf{Maj})[11]$ we obtain that rMLPs with 6 layers are strictly more c-interpretable than rMLPs with 36 layers. We generalize this observation in the following result.

**Proposition 12.** *Assume that the* $W(\mathsf{Maj})$-*hierarchy is strict. Then for every $t \geq 1$ we have that rMLPs with $3t + 3$ layers are strictly more c-interpretable than rMLPs with $9t + 27$ layers wrt. MCR.*

## 6 Discussion and concluding remarks

**Related work.** The need for model interpretability in machine learning has been heavily advocated during the last few years, with works covering theoretical and practical issues [3, 19, 23, 26, 27]. Nevertheless, a formal definition of interpretability has remained elusive [23]. In parallel, a related notion of interpretability has emerged from the field of *knowledge compilation* [9, 30, 31, 32, 33]. The intuition here is to construct a simpler and more interpretable model from a complex one. One can then study the simpler model to understand how the initial one makes predictions. Motivated by this, Darwiche and Hirth [8] use variations of the notion of sufficient reason to explore the interpretability of *Ordered BDDs* (OBDDs). The FBDDs that we consider in our work generalize OBDDs, and thus, our results for sufficient reasons over FBDDs can be seen as generalizations of the results in [8]. We consider FBDDs instead of OBDDs as FBDDs subsume decision trees, while OBDDs do not. We point out here that the notion of sufficient reason for a Boolean classifier is the same as the notion of *implicant* for a Boolean function, and that *minimal* sufficient reasons (with minimailty refering to subset-inclusion of the defined components) correspond to *prime implicants* [9]. We did not incorporate a study of minimal sufficient reasons (also called *PI-explanations*) to our work due to space constraints. In a contemporaneous work [24], Marques-Silva et al. study the task of enumerating the minimal sufficient reasons of naïve Bayes and linear classifiers. The queries COUNTCOMPLETIONS and CHECKSUFFICIENTREASON have already been studied for FBDDs in [9] (CHECKSUFFICIENTREASON under the name of *implicant check*). The query MINIMUMCHANGEREQUIRED is studied in [31] for OBDDs, where it is called *robustness*. Finally, there are papers exploring queries beyond the ones presented here [30, 31, 32], such as *monotonicity*, *unateness*, *bias detection*, *minimum cardinality explanations*, etc.

**Limitations.** Our framework provides a formal way of studying interpretability for classes of models, but still can be improved in several respects. One of them is the use of a more sophisticated complexity analysis that is not so much focused on the *worst case* complexity study propose here, but on identifying relevant parameters that characterize more precisely how difficult it is to interpret a particular class of models in practice. Also, in this paper we have focused on studying the local interpretability of models (why did the model make a certain prediction on a given input?), but one could also study their *global interpretability*, that is, making sense of the general relationships that a model has learned from the training data [27]. Our framework can easily be extended to the global setting by considering queries about models, independent of the input it receives. In order to avoid the difficulties of defining a general notion of interpretability [23], we have used explainability queries and their complexity as a formal proxy. Nonetheless, we do not claim that our notion of complexity-based interpretability is *the* definitive notion of interpretability. Indeed, most definitions of interpretability are directly related to humans in a subjective manner [5, 10, 25]. Our work is thus to be taken as a study of the correlation between a formal notion of interpretability and the folk wisdom regarding a subjective concept. Finally, even though the notion of complexity-based interpretability gives a precise way to compare models, our results show that it is still dependent on the particular set of queries that one picks. To achieve a more robust formalization of interpretability, one would then need to propose a more general approach that prescinds of specific queries. This is a challenging problem for future research.

## 7 Broader impact

Although interpretability as a subject may have a broad practical impact, our results in this paper are mostly theoretic, so we think that this work does not present any foreseeable societal consequences.

## Acknowledgments and Disclosure of Funding

Barceló and Pérez are funded by Fondecyt grant 1200967.

## Footnotes

[1]One has to be careful with this notation, however, as $\Sigma_2^p$ and #P are complexity classes for problems of different sort: the former being for decision problems, and the latter for counting problems. Although this issue can be solved by considering the class PP, we skip these technical details as they are not fundamental for the paper and can be found in most complexity theory textbooks, such as that of Arora and Barak [2].

[2]Recall that such an input consists of natural numbers (given in binary) $s_1, \ldots, s_n, k \in \mathbb{N}$, and a solution to it is a set $S \subseteq \{1, \ldots, n\}$ with $\sum_{i \in S} s_i \leq k$.

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
