[Supplementary Material]

# Appendix

The appendix contains the proofs for all the results presented in the main document. It is organized as follows:

## Appendix A.  Simulating Boolean formulas/circuits with MLPs

In this section we show that multilayer perceptrons can efficiently simulate arbitrary Boolean formulas. We will often use this result throughout the appendix to prove the hardness of our explainability queries over MLPs. In fact, and this will make the proof cleaner, we will show a slightly more general result: that MLPs can simulate arbitrary *Boolean circuits*. Formally, we show:

**Lemma 13.** *Given as input a Boolean circuit $C$, we can build in polynomial time an MLP $\mathcal{M}_C$ that is equivalent to $C$ as a Boolean function.*

*Proof.* We will proceed in three steps. The first step is to build from $C$ another equivalent circuit $C'$ that uses only what we call *relu gates*. A relu gate is a gate that, on input $\boldsymbol{x} = (x_1, \ldots, x_m) \in \mathbb{R}^m$, outputs $\mathrm{relu}(\langle \boldsymbol{w}, \boldsymbol{x} \rangle + b)$, for some rationals $\boldsymbol{w}_1, \ldots, \boldsymbol{w}_m, b$. Observe that these gates do not necessarily output $0$ or $1$, and so the circuit $C'$ might not be Boolean. However, we will ensure in the construction that the output of every relu gate in $C'$, when given Boolean inputs (i.e., $\boldsymbol{x} \in \{0,1\}^m$), is Boolean. This will imply that the circuit $C'$ is Boolean as well. To this end, it is enough to show how to simulate each original type of internal gate (NOT, OR, AND) by relu gates. We do so as follows:

- NOT-gate: simulated with a relu gate with only one weight of value $-1$ and a bias of $1$. Indeed, it is clear that for $x \in \{0,1\}$, we have that $\mathrm{relu}(-x+1) = \begin{cases} 1 & \text{if } x = 0 \\ 0 & \text{if } x = 1 \end{cases}$.

- AND-gate of in-degree $n$: simulated with a relu gate with $n$ weights, each of value $1$, and a bias of value $-(n-1)$. Indeed, it is clear that for $\boldsymbol{x} \in \{0,1\}^n$, we have that $\mathrm{relu}(\sum_{i=1}^n x_i - (n-1)) = \begin{cases} 1 & \text{if } \bigwedge_{i=1}^n x_i = 1 \\ 0 & \text{otherwise} \end{cases}$.

- OR-gate of in-degree $n$: we first rewrite the OR-gate with NOT- and AND-gates using De Morgan's laws, and then we use the last two items.

The second step is to build a circuit $C''$, again using only relu gates, that is equivalent to $C'$ and that is what we call *layerized*. This means that there exists a *leveling function* $l : C'' \to \mathbb{N}$ that

assigns to every gate of $C'$ a *level* such that (i) every variable gate is assigned level $0$, and (ii) for any wire $g \to g'$ (meaning that $g$ is an input to $g'$) in $C''$ we have that $l(g') = l(g) + 1$. To this end, let us call a relu gate that has a single input and weight $1$ and bias $0$ an *identity gate*, and observe then that the value of an identity gate is the same as the value of its only input, when this input is in $\{0, 1\}$. We will obtain $C''$ from $C'$ by inserting identity gates in between the gates of $C'$, which clearly does not change the Boolean function being computed. We can do so naïvely as follows. First, we initialize $l(g)$ to $0$ for all the variable gates $g$ of $C'$. We then iterate the following process: select a gate $g$ such that $l(g)$ is undefined and such that $l(g')$ is defined for every input $g'$ of $g$. Let $g_1', \ldots, g_m'$ be the inputs of $g$, and assume that $l(g_1') \leq \ldots \leq l(g_m')$. For every $1 \leq i \leq m$, we insert a line of $l(g_m') - l(g_i')$ identity gates in between $g_i'$ and $g$, and we set $l(g) \coloneqq l(g_m') + 1$, and we set the levels of the identity gates that we have inserted appropriately. It is clear that this construction can be done in polynomial time and that the resulting circuit $C''$ is layerized.

Finally, the last step is to transform $C''$ into an MLP $\mathcal{M}_C$ using only relu for the internal activation functions and the step function for the output layer (i.e., what we simply call "an MLP" in the paper), and that respects the structure given by our definition in Section 3.1 (i.e., where all neurons of a given layer are connected to all the neurons of the preceding layer). We first deal with having a step gate instead of a relu gate for the output. To achieve this, we create a fresh identity gate $g_0$, we set the output of $C''$ to be an input of $g_0$, and we set $g_0$ to be the new output gate of $C''$ (this does not change the Boolean function computed). We then replace $g_0$ by a *step gate* (which, we recall, on input $x \in \mathbb{R}$ outputs $0$ if $x < 0$ and $1$ otherwise) with a weight of $2$ and bias of $-1$, which again does not change the Boolean function computed; indeed, for $x \in \{0, 1\}$, we have that $\mathrm{step}(2x - 1) = \begin{cases} 1 & \text{if } x = 1 \\ 0 & \text{if } x = 0 \end{cases}$.

The level of $g_0$ is one plus the level of the previous output gate of $C''$. Therefore, to make $C''$ become a valid MLP, it is enough to do the following: for every gate $g$ of level $i$ and gate $g'$ of level $i + 1$, if $g$ and $g'$ are not connected in $C''$, we make $g$ be an input of $g'$ and we set the corresponding weight to $0$. This clearly does not change the function computed, and the obtained circuit can directly be regarded as an equivalent MLP $\mathcal{M}_C$. Since the whole construction can be performed in polynomial time, this concludes the proof. $\qquad\square$

## Appendix B. Proof of Proposition 5

In this section we prove Proposition 5. We recall its statement for the reader's convenience:

**Proposition 5.** *The* MINIMUMCHANGEREQUIRED *query is (1) in* PTIME *for FBDDs, (2) in* PTIME *for perceptrons, and (3)* NP*-complete for MLPs.*

We prove each item separately.

**Lemma 14.** *The* MINIMUMCHANGEREQUIRED *query can be solved in linear time for FBDDs.*

*Proof.* Let $(\mathcal{M}, \boldsymbol{x}, k)$ be an instance of MINIMUMCHANGEREQUIRED, where $\mathcal{M}$ is an FBDD. For every node $u$ in $\mathcal{M}$ we define $\mathcal{M}_u$ to be the FBDD obtained by restricting $\mathcal{M}$ to the nodes that are (forward-)reachable from $u$; in other words, $\mathcal{M}_u$ is the sub-FBDD rooted at $u$. Then, we define $\mathrm{mcr}_u(\boldsymbol{x})$ to be the minimum change required on $\boldsymbol{x}$ to obtain a classification under $\mathcal{M}_u$ that differs from $\mathcal{M}(\boldsymbol{x})$. More formally,

$$\mathrm{mcr}_u(\boldsymbol{x}) = \min\{k' \mid \text{there exists an instance } \boldsymbol{y} \text{ such that } d(\boldsymbol{x}, \boldsymbol{y}) = k' \text{ and } \mathcal{M}_u(\boldsymbol{y}) \neq \mathcal{M}(\boldsymbol{x})\},$$

with the convention that $\min \varnothing = \infty$. Observe that, (†) for an instance $\boldsymbol{y}$ minimizing $k'$ in this equality, since the FBDD $\mathcal{M}_u$ does not depend on the features associated to any node $u'$ from the root of $\mathcal{M}$ to $u$ excluded, we have that for any such node $\boldsymbol{y}_{u'} = \boldsymbol{x}_{u'}$ holds (otherwise $k'$ would not be minimized).[3] Let $r$ be the root of $\mathcal{M}$. Then, by definition we have that $(\mathcal{M}, \boldsymbol{x}, k)$ is a positive instance of MINIMUMCHANGEREQUIRED if and only $\mathrm{mcr}_r(\boldsymbol{x}) \leq k$. We now explain how we can compute all the values $\mathrm{mcr}_u(\boldsymbol{x})$ for every node $u$ of $\mathcal{M}$ in linear time.

By definition, if $u$ is a leaf labeled with true we have that $\mathcal{M}_u(\boldsymbol{y}) = 1$ for every $\boldsymbol{y}$, and thus if $\mathcal{M}(\boldsymbol{x}) = 0$ we get $\mathrm{mcr}_u(\boldsymbol{x}) = 0$, while if $\mathcal{M}(\boldsymbol{x}) = 1$ we get that $\mathrm{mcr}_u(\boldsymbol{x}) = \infty$. Analogously, if $u$ is a leaf labeled with false, then $\mathrm{mcr}_u(\boldsymbol{x})$ is equal to 0 if $\mathcal{M}(\boldsymbol{x}) = 1$ and to $\infty$ otherwise.

For the recursive case, we consider a non-leaf node $u$. Let $u_1$ be the node going along the edge labeled with 1 from $u$, and $u_0$ analogously. Using the notation $[x_u = a]$ to mean 1 if the feature of $\boldsymbol{x}$ indexed by the label of node $u$ has value $a \in \{0, 1\}$, and 0 otherwise, and the convention that $\infty + 1 = \infty$, we claim that:

$$\mathrm{mcr}_u(\boldsymbol{x}) = \min\Big([x_u = 1] + \mathrm{mcr}_{u_0}(\boldsymbol{x}), [x_u = 0] + \mathrm{mcr}_{u_1}(\boldsymbol{x})\Big)$$

Indeed, consider by inductive hypothesis that $\mathrm{mcr}_{u_0}(\boldsymbol{x})$ and $\mathrm{mcr}_{u_1}(\boldsymbol{x})$ have been properly calculated, and let us show that this equality holds. We prove both inequalities in turn:

- We show that $\mathrm{mcr}_u(\boldsymbol{x}) \leq \min\Big([x_u = 1] + \mathrm{mcr}_{u_0}(\boldsymbol{x}), [x_u = 0] + \mathrm{mcr}_{u_1}(\boldsymbol{x})\Big)$. It is enough to show that both $\mathrm{mcr}_u(\boldsymbol{x}) \leq [x_u = 1] + \mathrm{mcr}_{u_0}(\boldsymbol{x})$ and $\mathrm{mcr}_u(\boldsymbol{x}) \leq [x_u = 0] + \mathrm{mcr}_{u_1}(\boldsymbol{x})$ hold. We only show the first inequality, as the other one is similar. If $\mathrm{mcr}_{u_0}(\boldsymbol{x}) = \infty$ then clearly the inequality holds, hence let us assume that $\mathrm{mcr}_{u_0}(\boldsymbol{x}) = k' \in \mathbb{N}$. This means that there is an instance $\boldsymbol{y}'$ such that $d(\boldsymbol{x}, \boldsymbol{y}') = k'$ and such that $\mathcal{M}_{u_0}(\boldsymbol{y}') \neq \mathcal{M}(\boldsymbol{x})$. Furthermore, by the observation (†) we have that for any node $u'$ from the root of $\mathcal{M}$ to $u$ (included), we have $\boldsymbol{y}_{u'} = \boldsymbol{x}_{u'}$. Therefore, the instance $\boldsymbol{y}$ that is equal to $\boldsymbol{y}'$ but has value $\boldsymbol{y}_u = 0$ differs from $\boldsymbol{x}$ in exactly $k'' = [x_u = 1] + k'$, which implies that $\mathrm{mcr}_u(\boldsymbol{x}) \leq [x_u = 1] + \mathrm{mcr}_{u_0}(\boldsymbol{x})$. Hence, the first inequality is proven.

- We show that $\mathrm{mcr}_u(\boldsymbol{x}) \geq \min\Big([x_u = 1] + \mathrm{mcr}_{u_0}(\boldsymbol{x}), [x_u = 0] + \mathrm{mcr}_{u_1}(\boldsymbol{x})\Big)$. First, assume that both $\mathrm{mcr}_{u_0}(\boldsymbol{x})$ and $\mathrm{mcr}_{u_1}(\boldsymbol{x})$ are equal to $\infty$. This means that every path in both $\mathcal{M}_{u_0}$ and $\mathcal{M}_{u_1}$ leads to a leaf with the same classification as $\mathcal{M}(\boldsymbol{x})$. Then, as every path from $u$ goes either through $u_0$ or through $u_1$, it must be that every path from $u$ leads to a leaf with the same classification as $\mathcal{M}(\boldsymbol{x})$, and thus $\mathrm{mcr}_u(\boldsymbol{x}) = \infty$, and so the inequality holds. Therefore, we can assume that one of $\mathrm{mcr}_{u_0}(\boldsymbol{x})$ or $\mathrm{mcr}_{u_1}(\boldsymbol{x})$ is finite. Let us assume without loss of generality that $(\star)$ $\min\Big([x_u = 1] + \mathrm{mcr}_{u_0}(\boldsymbol{x}), [x_u = 0] + \mathrm{mcr}_{u_1}(\boldsymbol{x})\Big) = [x_u = 1] + \mathrm{mcr}_{u_0}(\boldsymbol{x}) \in \mathbb{N}$ (the other case being similar). Let us now assume, by way of contradiction, that the inequality does not hold, that is, we have that (††) $\mathrm{mcr}_u(\boldsymbol{x}) < [x_u = 1] + \mathrm{mcr}_{u_0}(\boldsymbol{x})$, and let $\boldsymbol{y}$ be an instance such that $\mathcal{M}_u(\boldsymbol{y}) \neq \mathcal{M}_u(\boldsymbol{x})$ and $\mathrm{d}(\boldsymbol{x}, \boldsymbol{y}) = \mathrm{mcr}_u(\boldsymbol{x})$. Thanks to $(\star)$, we can assume wlog that $\boldsymbol{y}_u = 0$. But then we would have that $\mathrm{mcr}_{u_0}(\boldsymbol{x}) \leq \mathrm{mcr}_u(\boldsymbol{x}) - [x_u = 1]$, which contradicts (††). Hence, the second inequality is proven.

It is clear that the recursive function $\mathrm{mcr}$ can be computed bottom-up in linear time, thus concluding the proof. $\square$

**Lemma 15.** *The* MINIMUMCHANGEREQUIRED *query can be solved in linear time for perceptrons.*

*Proof.* Let $(\mathcal{M} = (\boldsymbol{w}, b), \boldsymbol{x}, k)$ be an instance of the problem, and let us assume without loss of generality that $\mathcal{M}(\boldsymbol{x}) = 1$, as the other case is analogous. For each feature $i$ of $\boldsymbol{x}$ we define its importance $s(i)$ as $w_i$ if $x_i = 1$ and $-w_i$ otherwise. Intuitively, $s$ represents how good it is to keep a certain feature in order to maintain the verdict of the model. We now assume that $\boldsymbol{x}$ and $\boldsymbol{w}$ have been sorted in decreasing order of score $s$ (paying the cost of a sorting procedure). For example, if originally $\boldsymbol{w} = (3, -5, -2)$ and $\boldsymbol{x} = (1, 0, 1)$, then after the sorting procedure we have $\boldsymbol{w} = (-5, 3, -2)$ and $\boldsymbol{x} = (0, 1, 1)$. This sorting procedure has cost $O(|\mathcal{M}|)$ as it is a classical problem of sorting strings whose total length add up to $\mathcal{M}$ and can be carried with a variant of Bucketsort [7]. As a result, for every pair $1 \leq i < j \leq n$ we have that $s(i) \geq s(j)$. Let $k'$ be the largest integer no greater than $k$ such that $s(k') > 0$ and then define $\boldsymbol{x}'$ as the instance that differs from $\boldsymbol{x}$ exactly on the first $k'$ features. We claim that $\mathcal{M}(\boldsymbol{x}') \neq \mathcal{M}(\boldsymbol{x})$ if and only if $(\mathcal{M}, \boldsymbol{x}, k)$ is a positive instance of MINIMUMCHANGEREQUIRED. The forward direction follows from the fact that $k' \leq k$. For the backward direction, assume that $(\mathcal{M}, \boldsymbol{x}, k)$ is a positive instance of MINIMUMCHANGEREQUIRED. This implies that there is an instance $\boldsymbol{y}$ that differs from $\boldsymbol{x}$ in at

most $k$ features, and for which $\mathcal{M}(\boldsymbol{y}) = 0$. If $\boldsymbol{y} = \boldsymbol{x}'$, then we are immediately done, so we can safely assume this is not the case.

We then define, for any instance $\boldsymbol{y}$ of $\mathcal{M}$ the function $v(\boldsymbol{y}) = \langle \boldsymbol{w}, \boldsymbol{y} \rangle$. Note that an instance $\boldsymbol{y}$ of $\mathcal{M}$ is positive if and only if $v(\boldsymbol{y}) \geq -b$. Then, since we have that $\mathcal{M}(\boldsymbol{y}) = 0$, it holds that $v(\boldsymbol{y}) < -b$. We now claim that $v(\boldsymbol{x}') \leq v(\boldsymbol{y})$:

**Claim 16.** *For every instance $\boldsymbol{y}$ such that $d(\boldsymbol{y}, \boldsymbol{x}) \leq k$ and $\mathcal{M}(\boldsymbol{y}) \neq \mathcal{M}(\boldsymbol{x})$, it must hold that $v(\boldsymbol{x}') \leq v(\boldsymbol{y})$.*

*Proof.* For an instance $\boldsymbol{z}$, let us write $C_{\boldsymbol{z}}$ for the set of features for which $\boldsymbol{z}$ differs from $\boldsymbol{x}$. We then have on the one hand

$$v(\boldsymbol{x}') = \sum_{i \in C_{\boldsymbol{x}'} \setminus C_{\boldsymbol{y}}} (1 - x_i) w_i + \sum_{i \in C_{\boldsymbol{y}} \cap C_{\boldsymbol{x}'}} (1 - x_i) w_i + \sum_{i \notin C_{\boldsymbol{x}'} \cup C_{\boldsymbol{y}}} x_i w_i + \sum_{i \in C_{\boldsymbol{y}} \setminus C_{\boldsymbol{x}'}} x_i w_i$$

and on the other hand

$$v(\boldsymbol{y}) = \sum_{i \in C_{\boldsymbol{y}} \setminus C_{\boldsymbol{x}'}} (1 - x_i) \boldsymbol{w}_i + \sum_{i \in C_{\boldsymbol{y}} \cap C_{\boldsymbol{x}'}} (1 - x_i) w_i + \sum_{i \notin C_{\boldsymbol{x}'} \cup C_{\boldsymbol{y}}} x_i w_i + \sum_{i \in C_{\boldsymbol{x}'} \setminus C_{\boldsymbol{y}}} x_i w_i$$

As the two middle terms are shared, we only need to prove that

$$\sum_{i \in C_{\boldsymbol{x}'} \setminus C_{\boldsymbol{y}}} (1 - x_i) w_i + \sum_{i \in C_{\boldsymbol{y}} \setminus C_{\boldsymbol{x}'}} x_i w_i \leq \sum_{i \in C_{\boldsymbol{y}} \setminus C_{\boldsymbol{x}'}} (1 - x_i) w_i + \sum_{i \in C_{\boldsymbol{x}'} \setminus C_{\boldsymbol{y}}} x_i w_i$$

which is equivalent to proving that

$$\sum_{i \in C_{\boldsymbol{x}'} \setminus C_{\boldsymbol{y}}, x_i = 0} w_i + \sum_{i \in C_{\boldsymbol{y}} \setminus C_{\boldsymbol{x}'}, x_i = 1} w_i \leq \sum_{i \in C_{\boldsymbol{y}} \setminus C_{\boldsymbol{x}'}, x_i = 0} w_i + \sum_{i \in C_{\boldsymbol{x}'} \setminus C_{\boldsymbol{y}}, x_i = 1} w_i$$

and by using the definition of importance, equivalent to

$$\sum_{i \in C_{\boldsymbol{x}'} \setminus C_{\boldsymbol{y}}, x_i = 0} -s(i) + \sum_{i \in C_{\boldsymbol{y}} \setminus C_{\boldsymbol{x}'}, x_i = 1} s(i) \leq \sum_{i \in C_{\boldsymbol{y}} \setminus C_{\boldsymbol{x}'}, x_i = 0} -s(i) + \sum_{i \in C_{\boldsymbol{x}'} \setminus C_{\boldsymbol{y}}, x_i = 1} s(i)$$

which can be rearranged into

$$\sum_{i \in C_{\boldsymbol{y}} \setminus C_{\boldsymbol{x}'}} s(i) \leq \sum_{i \in C_{\boldsymbol{x}'} \setminus C_{\boldsymbol{y}}} s(i)$$

But this inequality must hold as $C_{\boldsymbol{x}}'$ is by definition the set $C$ of features of size at most $k$ that maximizes $\sum_{i \in C} s(i)$. $\qquad \square$

Because of the claim, and the fact that $v(\boldsymbol{y}) < -b$ we conclude that $v(\boldsymbol{x}') < -b$, and thus $\mathcal{M}(\boldsymbol{x}') \neq \mathcal{M}(\boldsymbol{x})$. This concludes the backward direction, and thus, the fact that checking whether $\mathcal{M}(\boldsymbol{x}') \neq \mathcal{M}(\boldsymbol{x})$ is enough to solve the entire problem. Since checking this can be done in linear time, constructing $\boldsymbol{x}'$ is the most expensive part of the process, which can effectively be done in time $O(|\mathcal{M}|)$. This concludes the proof of the lemma. $\qquad \square$

**Lemma 17.** *The* MINIMUMCHANGEREQUIRED *query is* NP-*complete for MLPs.*

*Proof.* Membership is easy to see, it is enough to non-deterministically guess an instance $\boldsymbol{y}$ and check that $d(\boldsymbol{x}, \boldsymbol{y}) \leq k$ and $\mathcal{M}(\boldsymbol{x}) \neq \mathcal{M}(\boldsymbol{y})$.

In order to prove hardness, we reduce from VERTEX COVER. Given an undirected graph $G = (V, E)$ and an integer $k$, the VERTEX COVER problem consists in deciding whether there is a subset $S \subseteq V$ of at most $k$ vertices such that every edge of $G$ touches a vertex in $S$. Let $(G = (V, E), k)$ be an instance of VERTEX COVER, and let $n$ denote $|V|$. Based on $G$, we build a formula $\varphi_G$, where propositional variables correspond to vertices of $G$.

$$\varphi_G = \bigwedge_{(u,v) \in E} (x_u \vee x_v)$$

It is clear that the satisfying assignments of $\varphi_G$ correspond to the vertex covers of $G$, and furthermore, that a satisfying assignment of Hamming weight $k$ (number of variables assigned to 1) corresponds to a vertex cover of size $k$.

Moreover, we can safely assume that there is at least 1 edge in $G$, as otherwise the instance would be trivial, and a constant size positive instance of MCR would finish the reduction. This implies in turn, that we can assume that assigning every variable to 0 does not satisfy $\varphi_G$.

We now build an MLP $\mathcal{M}_\varphi$ from $\varphi_G$, using Lemma 13. We claim that the instance $(\mathcal{M}_\varphi, 0^n, k)$ is a positive instance of MINIMUMCHANGEREQUIRED if and only if $(G, k)$ is a positive instance of VERTEX COVER.

Indeed, $0^n$ is a negative instance of $\mathcal{M}_\varphi$, as assigning every variable to 0 does not satisfy $\varphi_G$. Moreover a positive instance of weight $k$ for $\mathcal{M}_\varphi$ corresponds to a satisfying assignment of weight $k$ for $\varphi_G$, which in turn corresponds to a vertex cover of size $k$ for $G$. This is enough to conclude conclude the proof, recalling that both the construction of $\varphi_G$ and $\mathcal{M}_\varphi$ take polynomial time. $\qquad\square$

## Appendix C.  Proof of Proposition 6

In this section we prove Proposition 6, whose statement we recall here:

**Proposition 6.** *The* MINIMUMSUFFICIENTREASON *query is (1)* NP-*complete for FBDDs (and hardness holds already for decision trees), (2) in* PTIME *for perceptrons, and (3)* $\Sigma_2^p$-*complete for MLPs.*

Again, we prove each claim separately.

**Lemma 18.** *The* MINIMUMSUFFICIENTREASON *query is* NP-*complete for FBDDs, and hardness holds already for decision trees.*

*Proof.* Membership in NP is clear, it suffices to guess the instance $\boldsymbol{y}$ and check both that it has less than $k$ defined components and that is a sufficient reason for $\boldsymbol{x}$, which can be done thanks to Lemma 23. We will prove that hardness holds already for the particular case of decision trees, and when the input instance $\boldsymbol{x}$ is positive. Hardness of this particular setting implies of course the hardness of the general problem. In order to do so, we will reduce from the problem of determining whether a directed acyclic graph has a dominating set of size at most $k$, which we abbreviate as DOM-DAG. Recall that in a directed graph $G = (V, E)$, a subset of vertices $D \subseteq V$ is said to be dominating if every vertex in $V \setminus D$ has an incoming edge from a vertex in $D$. The problem of DOM-DAG is shown to be NP-complete in [22].

An illustration of the reduction is presented in Figure 2. Let $(G = (V, E), k)$ be an instance of DOM-DAG, and let us define $n := |V|$. We start by computing in polynomial time a topological ordering $\varphi = \varphi_1, \ldots, \varphi_n$ of $G$. Next, we will create an instance $(\mathcal{T}, \boldsymbol{x}, k)$ of $k$-SUFFICIENTREASON such that there is a sufficient reason of size at most $k$ for $\boldsymbol{x}$ under the decision tree $\mathcal{T}$ if and only if $G$ has a dominating set of size at most $k$. We create the decision tree $\mathcal{T}$, of dimension $n$, in 2 steps.

1. Create nodes $v_1, \ldots, v_n$, where node $v_i$ is labeled with $\varphi_i$ The node $v_n$ will be the root of $\mathcal{T}$, and for $2 \leq i \leq n$, connect $v_i$ to $v_{i-1}$ with an edge labeled with 1. Node $v_1$ is connected to a leaf labeled true through an edge labeled with 1. We will denote the path created in this step as $\pi$.

2. For every vertex $\varphi_i$ create a decision tree $\mathcal{T}_i$ equivalent to the boolean formula

$$\mathcal{F}_i = \bigvee_{(\varphi_j, \varphi_i) \in E} \varphi_j$$

   and create an edge from $v_i$ to the root of $\mathcal{T}_i$ labeled with 0. If $\mathcal{F}_i$ happens to be the empty formula, $\mathcal{T}_i$ is defined as false. Note that the nodes introduced by this step are all naturally associated with vertices of $G$.

Step 2 takes polynomial time because boolean formulas in 1-DNF can easily be transformed into a decision tree in linear time.

(a) Example of an input DAG. Nodes 2 and 5, corresponding to the minimum dominating set of $G$ are emphasized.

(b) A topological ordering $\varphi$ of $G$.

(c) Resulting decision tree $\mathcal{T}$. Edges to the left of a node are always labeled with 0, and edges to the right with 1. The leaves are not depicted for clarity, but: if a node has no right child in the picture, then its right child is true, and if it has no left child then its left child is false. Note that in every diagonal there is an emphasized node, which is either 2 or 5, implying the partial instance $(\bot, 1, \bot, \bot, 1, \bot)$ is a sufficient reason for the instance $\boldsymbol{x} = (1, 1, 1, 1, 1, 1)$.

Figure 2: Illustration of the reduction from DOM-DAG to $k$-SUFFICIENTREASON over decision trees, for an example graph of 6 nodes.

We now check that $\mathcal{T}$ is a decision tree. Since $\mathcal{T}$ has a tree structure, it is enough to check that for every path from the root to a leaf there are no two nodes on the path that have the same label (i.e., to check that $\mathcal{T}$ is a valid FBDD). Note that any path from the root $v_n$ to a leaf goes first to a certain node $v_i$ in $\pi$, from where it either takes an edge labeled with 0, in case $i \neq 1$ or it simply goes to a leaf otherwise. In case $i = 1$, the path from the root goes exactly through $v_n, v_{n-1}, \ldots, v_1$, which all have different labels. In case $i \neq 1$, the path includes (i) nodes with labels $\varphi_n, \varphi_{n-1}, \ldots, \varphi_i$, and (ii) a subpath inside $\mathcal{T}_i$. It is clear that all the labels in (i) are different. And as by construction $\mathcal{T}_i$ is a decision tree, no two nodes inside (ii) can have the same label. It remains to check that no node in (i) can have the same label of a node in (ii). To see this, consider that all the vertices of $G$ associated to the nodes in (ii) have edges to $\varphi_i$ in $G$, and thus come before $\varphi_i$ in the topological order. But (i) is composed precisely by $\varphi_i$ and the nodes who come after it in the topological ordering, so (i) and (ii) have empty intersection.

Let $\boldsymbol{x} = 1^n$ be the vector of $n$ ones. We claim that $(\mathcal{T}, \boldsymbol{x}, k)$ is a yes-instance of $k$-SUFFICIENTREASON if and only if $(G, k)$ is a yes-instance of DOM-DAG.

**Forward direction.** Consider that there is a sufficient reason $\boldsymbol{y}$ for $\boldsymbol{x}$ under $\mathcal{T}$ of size at most $k$. As $\boldsymbol{x}$ contains only 1s, $\boldsymbol{y}$ must contain only 1s and $\bot$s. Consider the set $S$ of components $i$ where $y_i = 1$. Recalling that every vertex of $G$ is canonically associated with a feature of $\mathcal{T}$, we will denote $D_S$ to the set of vertices of $G$ that are associated with the features in $S$. It is clear that $|D_S| \leq k$. We now prove that $D_S$ is a dominating set of $G$. First, in case $D_S = V$, we are trivially done. We know assume $D_S \neq V$. Consider a vertex $v \in V \setminus D_S$, corresponding to $\varphi_i$ in the topological ordering, and define $\boldsymbol{z}$ as the completion of $\boldsymbol{y}$ where the features $\varphi_j$ such that $j > i$, are set to 1, and all other features that are undefined by $\boldsymbol{y}$ are set to 0. By hypothesis, $\boldsymbol{z}$ must be a positive instance, and so its path on $\mathcal{T}$ must end in a leaf labeled with true. Note that the path of $\boldsymbol{z}$ in $\mathcal{T}$ necessarily takes the path $\pi$ created in Step 1 of the construction, up to the node $v_i$, and then enters its subtree $\mathcal{T}_i$. Let $t$ be the node of $\mathcal{T}_i$ whose leaf labeled with true ends the path of $\boldsymbol{z}$ in $\mathcal{T}$, and $\varphi_k$ its label and associated vertex in $G$. As feature $t$ is set to 1, we must have either $\varphi_k \in D_S$ (in case $t$ is 1 because of $\boldsymbol{y}$) or $k > i$ (in case $t$ is 1 by the construction of completion $\boldsymbol{z}$). However, the second case is not actually possible, as if $k > i$, that means $v_k$ comes before $v_i$ in path $\pi$, and thus the path of $\boldsymbol{z}$ in $\mathcal{T}$ passes through $v_k$, which has label $\varphi_k$ before passing through $v_i$. But the path of $\boldsymbol{z}$ in $\mathcal{T}$ passes through $t$ before ending, which also has label $\varphi_k$. This contradicts the already proven fact that $\mathcal{T}$ is a decision tree. We can therefore assume that $\varphi_k$ belongs to $D_S$. Then, as $t$ is a node of $\mathcal{T}_i$, there must be an edge $(\varphi_k, \varphi_i)$ in $E$ because of the way $\mathcal{T}_i$ is constructed. But this means that vertex $v \in V \setminus D_S$ has an edge coming from $\varphi_k \in D_S$, and so $v$ is effectively dominated by the set $D_S$. As this holds for every $v \in V \setminus D_S$, we conclude that $D_S$ is indeed a dominating set of $G$.

**Backward Direction.** Consider that there is a dominating set $D \subseteq V$ of size at most $k$. Let $S_D$ be the set of features associated with $D$. We claim that the partial instance $\boldsymbol{y}$ that has 1 in the features that belong to $S_D$, and is undefined elsewhere, is a sufficient reason for $\boldsymbol{x}$, and by construction its size is at most $k$. Consider an arbitrary completion $\boldsymbol{z}$ of $\boldsymbol{y}$, we need to show that $\boldsymbol{z}$ is a positive instance of $\mathcal{T}$. For $\boldsymbol{z}$ not to be a positive instance, its path on $\mathcal{T}$ would have to reach a leaf labeled with false. This can only happen by either taking the edge labeled with 0 from $v_1$ (the last node in path $\pi$ built in the construction), or inside a subtree $\mathcal{T}_i$, corresponding to a node $v_i$ whose associated feature in $\boldsymbol{z}$ is set to 0. We show that neither can happen. For the first case, every dominating set must include $\varphi_1$, the vertex in $G$ associated with $v_1$, as it is the first element in the topological ordering of $G$, and thus it must has in-degree 0, which implies $\varphi_1 \in D$. Therefore, it is not possible to take the edge labeled with 0 from $v_1$. On the other hand, suppose the path of $\boldsymbol{z}$ in $\mathcal{T}_i$ ends in a leaf labeled with false. Then, by construction of $\mathcal{T}_i$, there is no vertex $\varphi_j$ such that $(\varphi_j, \varphi_i) \in E$ whose associated feature is set to 1 in $\boldsymbol{z}$. But as $D$ is a dominating set, either there is indeed a $\varphi_j \in D$ such that $(\varphi_j, \varphi_i) \in E$ or $\varphi_i \in D$. The first case is in direct contradiction with the previous statement, as $\varphi_j \in D$ implies, by our construction of $\boldsymbol{y}$ that the feature associated with $\varphi_j$ is set to 1. The second case also creates a contradiction, as if $\varphi_i \in D$, then by construction $\boldsymbol{y}$ would have a 1 in the feature $v_i$ associated to $\varphi_i$, which contradicts the assumption of the path of $\boldsymbol{z}$ entering $\mathcal{T}_i$. □

**Lemma 19.** *The* MINIMUMSUFFICIENTREASON *query is in* PTIME *for perceptrons.*

*Proof.* Let $(\mathcal{M} = (\boldsymbol{w}, b), \boldsymbol{x}, k)$ be an instance of the problem, and let $d$ be the dimension of the perceptron. We will assume without loss of generality that $\mathcal{M}(\boldsymbol{x}) = 1$. In this proof, what we call a *minimum sufficient reason for $\boldsymbol{x}$* is a sufficient reason for $\boldsymbol{x}$ that has the least number of components being defined. We show a greedy algorithm that computes a minimum sufficient reason for $\boldsymbol{x}$ under $\mathcal{M}$ in time $O(|\mathcal{M}|)$. For each feature $i$ of $\boldsymbol{x}$ we define its importance $s(i)$ as $w_i$ if $x_i = 1$ and $-w_i$ otherwise (just as we did in the proof of Lemma 15), and its *penalty* $p(i)$ as $\min(0, w_i)$. Intuitively, $s$ represents how good it is for a partial instance to be defined in a given feature, and $p$ represents the penalty or cost that a partial instance incurs by not being defined in a given feature. We now assume that $\boldsymbol{x}$ and $\boldsymbol{w}$ have been sorted in decreasing order of score $s$. For example, if originally $\boldsymbol{w} = (3, -5, -2)$ and $\boldsymbol{x} = (1, 0, 1)$, then after the sorting procedure we have $\boldsymbol{w} = (-5, 3, -2)$ and $\boldsymbol{x} = (0, 1, 1)$. We now define a function $\psi$ that takes any partial instance $\boldsymbol{y}$ as input and outputs the worst possible value for a completion of $\boldsymbol{y}$:

$$\psi(\boldsymbol{y}) := \min_{\boldsymbol{z}: \boldsymbol{z} \text{ is a completion of } \boldsymbol{y}} \langle \boldsymbol{w}, \boldsymbol{z} \rangle = \sum_{y_i \neq \bot} w_i y_i + \sum_{y_i = \bot} p(i).$$

The second equality is easy to see based on the definition of the function $p$, and the definition of $\psi$ implies that $\psi(\boldsymbol{y}) \geq -b$ exactly when $\boldsymbol{y}$ is a sufficient reason. For $1 \leq l \leq d$, we define $\boldsymbol{y}^l$ as the partial instance of $\boldsymbol{x}$ such that $y_i^l$ is equal to $x_i$ if $i \leq l$ and to $\bot$ otherwise. In simple terms, $\boldsymbol{y}^l$ is the partial instance obtained by taking the first $l$ features of $\boldsymbol{x}$; continuing our example with $\boldsymbol{x} = (0, 1, 1)$, we have for instance $\boldsymbol{y}^2 = (0, 1, \bot)$. Let $j$ be the minimum index such that $\psi(\boldsymbol{y}^j) \geq -b$. Such an index always exists, because, since $\boldsymbol{x}$ is a positive instance, taking $j = d$ is always a valid index. Note that $j$ can be computed in linear time.

We now prove that (†) the partial instance $\boldsymbol{y}^j$ is a minimum sufficient reason for $\boldsymbol{x}$. By definition we have that $\psi(\boldsymbol{y}^j) \geq -b$, so $\boldsymbol{y}^j$ is indeed a sufficient reason for $\boldsymbol{x}$. We now need to show that $\boldsymbol{y}^j$ is minimum. Assume, seeking for a contradiction, that there is a sufficient reason $\boldsymbol{y}'$ of $\boldsymbol{x}$ with strictly less components defined than $\boldsymbol{y}^j$; clearly we can assume without loss of generality that $\boldsymbol{y}'$ has exactly $j - 1$ components defined. We will now show that (⋆) $\boldsymbol{y}^{j-1}$ is a also a sufficient reason for $\boldsymbol{x}$, which is a contradiction since $j$ was assumed to be the minimal index such that $\boldsymbol{y}^j$ is a sufficient reason of $\boldsymbol{x}$, hence proving (†). If $\boldsymbol{y}' = \boldsymbol{y}^{j-1}$, we have that (⋆) is trivially true. Otherwise, and considering that $\boldsymbol{y}'$ and $\boldsymbol{y}^{j-1}$ have the same size, and that $\boldsymbol{y}^{j-1}$ is defined exactly on the first $j - 1$ features, there must be at least a pair of features $(m, n)$, with $m \leq j - 1 < n$, such that $\boldsymbol{y}^{j-1}$ is defined at feature $m$ and $\boldsymbol{y}'$ is not, and on the other hand $\boldsymbol{y}'$ is defined at feature $n$ whereas $\boldsymbol{y}^{j-1}$ is not. In order to finish the proof of (⋆), we will prove a simpler claim that will help us conclude.

**Claim 20.** *Assume that there is a pair of features $(m, n)$ with $m \leq j - 1 < n$ such that $y_m' = \bot, y_m^{j-1} \neq \bot$ and $y_n' \neq \bot, y_n^{j-1} = \bot$, and let $\boldsymbol{y}^*$ be the resulting partial instance that is equal to $\boldsymbol{y}'$ except that $y_m^* := y_m^{j-1}$ and $y_n^* := \bot$. Then we have that $\psi(\boldsymbol{y}^*) \geq \psi(\boldsymbol{y}')$.*

*Proof of Claim 20.* By definition, $\psi(\boldsymbol{y}^*) - \psi(\boldsymbol{y}') = p(n) - p(m) + w_m y_m^{j-1} - w_n y_n' = p(n) - p(m) + w_m x_m - w_n x_n$. But because the features in $\boldsymbol{y}^{j-1}$ are sorted in decreasing order of score, it must hold that $s(m) \geq s(n)$. Using this last inequality and reasoning by cases on the values $x_m, x_n$ and on the signs of $w_m, w_n$, one can tediously check that $\psi(\boldsymbol{y}^*) - \psi(\boldsymbol{y}') \geq 0$ and thus $\psi(\boldsymbol{y}^*) \geq \psi(\boldsymbol{y}')$. $\qquad\square$

We now continue with the proof of $(\star)$. As a result of Claim 20, one can iteratively modify $\boldsymbol{y}'$ until it becomes equal to $\boldsymbol{y}^{j-1}$ in such a way that the value of $\psi$ is never decreased along the process, implying therefore that $\psi(\boldsymbol{y}^{j-1}) \geq \psi(\boldsymbol{y}')$. But $\psi(\boldsymbol{y}') \geq -b$, because $\boldsymbol{y}'$ is assumed to be a sufficient reason, hence we have that $\psi(\boldsymbol{y}^{j-1}) \geq -b$, implying that $\boldsymbol{y}^{j-1}$ is a sufficient reason for $\boldsymbol{x}$, and thus concluding the proof of $(\star)$. Therefore, $(\dagger)$ is proven, and since $\boldsymbol{y}^j$ can clearly be computed in polynomial time (in fact, the runtime of the whole procedure is dominated by the sorting subroutine, which again has cost $O(|\mathcal{M}|)$ as it is a classical problem of sorting strings whose total length add up to $|\mathcal{M}|$ and can be carried with a variant of Bucketsort [7]), this finishes the proof of the lemma; indeed, we can output YES if $j \leq k$ and NO otherwise. $\qquad\square$

**Lemma 21.** *The* MINIMUMSUFFICIENTREASON *query is* $\Sigma_2^p$*-complete for MLPs.*

*Proof.* Membership in $\Sigma_2^p$ is clear, as one can non-deterministically guess the value of the $k$ features that would make for a sufficient reason, and then use an oracle in co-NP to verify that no completion of that guess has a different classification. To show hardness, we will reduce from the problem SHORTEST IMPLICANT CORE, defined and proven to be $\Sigma_2^p$-hard by Umans [34, Theorem 1]. First, we need a few definitions in order to present this problem. A formula in *disjunctive normal form* (DNF) is a Boolean formula of the form $\varphi = t_1 \vee t_2 \vee \ldots \vee t_n$, where each *term* $t_i$ is a conjunction of literals (a literal being a variable of the negation thereof). An *implicant* for $\phi$ is a partial assignment of the variables of $\phi$ such that any extension to a full assignment makes the formula evaluate to true; note that we can equivalently see an implicant of $\phi$ as what we call a sufficient reason of $\phi$. For a partial assigment $C$ of the variables and for a set of literals $t$ (or conjunction of literals $t$), we write $C \subseteq t$ when for every variable $x$, if $x \in t$ then $C(x) = 1$ and if $\neg x \in t$ then $C(x) = 0$ and $C(x)$ is undefined otherwise. An instance of SHORTEST IMPLICANT CORE then consists of a DNF formula $\varphi = t_1 \vee t_2 \vee \ldots \vee t_n$, together with an integer $k$. Such an instance is positive for SHORTEST IMPLICANT CORE when there is an implicant $C$ for $\varphi$ such that $C \subseteq t_n$.[4] Note that the SHORTEST IMPLICANT CORE is closer to the problem at hand than the general SHORTEST IMPLICANT problem, as (minimum) sufficient reasons of an instance $\boldsymbol{x}$ can only induce literals according to $\boldsymbol{x}$, in a similar fashion of implicants that can only induce literals according to the core $t_n$.

**A reduction that does not work, and how to fix it on an example.** In order to convey the main intuition, we start by presenting a first tentative of a reduction that does not work. Thanks to Lemma 13 we know that it is possible to build an MLP $\mathcal{M}_\varphi$ equivalent to $\varphi$. However, doing so directly creates a problem: we would need to find a convenient instance $\boldsymbol{x}$ such that $(\varphi, k) \in$ SHORTEST IMPLICANT CORE if and only if $(\mathcal{M}_\varphi, \boldsymbol{x}, k) \in k\text{-}$SUFFICIENTREASON. A natural idea is to consider $t_n$ as a candidate for $\boldsymbol{x}$, but the issue is that $t_n$ does not necessarily include every variable. The next natural idea is to try with $\boldsymbol{x}$ being an arbitrary completion of $t_n$ (interpreting $t_n$ as the partial instance that is uniquely defined by its satisfying assignment). This approach fails because there could be a sufficient reason of size at most $k$ for such an $\boldsymbol{x}$ that relies on features (variables) that are not in $t_n$. We illustrate this with an example for $n = 4$.

$$\varphi := x_1 \overline{x_5} \vee \overline{x_2}\,\overline{x_6} \vee x_3 x_6 \vee \overline{x_1}\,\overline{x_2} x_4 \vee \underbrace{x_1 x_3 x_5}_{t_4}$$

While it can be checked that $(\varphi, 2) \notin$ SHORTEST IMPLICANT CORE, we have that $(\mathcal{M}_\varphi, (1, 0, 1, 0, 1, 1), 2)$ is in fact a positive instance of $k$-SUFFICIENTREASON, as the partial instance that assigns 1 to $x_3$ and $x_6$ and is undefined for the rest of the features, is a sufficient reason of size 2 for $\boldsymbol{x}$. The issue is that we are allowing $x_6$ to be part of the sufficient reason for $\boldsymbol{x}$ even though $x_6 \notin t_4$. We can avoid this from happening by splitting each variable that is not in $t_n$, such as $x_6$, into $k + 1$ variables, in such a way that defining the value of $x_6$ would force us to define the

value of all the $k + 1$ variables, which is of course unaffordable. Continuing with the example, we build the formula $\varphi'$ as follows:

$$\varphi' := \bigwedge_{i=1}^{3} \left( x_1 \overline{x_5} \vee \overline{x_2^i}\, \overline{x_6^i} \vee x_3 x_6^i \vee \overline{x_1}\, \overline{x_2^i} x_4^i \vee x_1 x_3 x_5 \right)$$

Now we can simply take $(\mathcal{M}_{\varphi'}, \boldsymbol{x}, 1)$ where $\boldsymbol{x}$ is an arbitrary completion of $t_4$ over the new set of variables, for example, one that assigns $1$ to the features $1, 3$ and $5$, and $0$ to all other features (variables). Note that $\varphi'$ is not a DNF anymore, but this is not a problem, since we only need to compute $\mathcal{M}_{\varphi'}$. It is then easy to check that this instance is equivalent to the original input instance.

**The reduction.** We now present the correct reduction and prove that it works. Let $(\varphi, k)$ be an instance of SHORTEST IMPLICANT CORE. Let $X_c$ be the set of variables that are not mentioned in $t_n$. We split every variable $x_j \in X_c$ into $k + 1$ variables $x_j^1, \ldots x_j^{k+1}$ and for each $i \in \{1, \ldots, k+1\}$ we build $\varphi^{(i)}$ by replacing every occurrence of a variable $x_j$, that belongs to $X_c$, by $x_j^i$. Finally we define $\varphi'$ as the conjunction of all the $\varphi^{(i)}$. That is,

$$\varphi^{(i)} := \varphi[x_j \to x_j^i, \text{ for all } x_j \in X_c] \tag{2}$$

$$\varphi' := \bigwedge_{i=1}^{k+1} \varphi^{(i)} \tag{3}$$

Observe that any meaningful instance of SHORTEST IMPLICANT CORE has $k < |t_n|$, so we can safely assume that $k$ is given in unary, making this construction polynomial.

We then use Lemma 13 to build an MLP $\mathcal{M}_{\varphi'}$ from $\varphi'$, in polynomial time. The features of this model correspond naturally to the variables of $\varphi'$, and thus we refer to both features and variables without distinction. Let $\boldsymbol{y}$ be the instance that assigns $1$ to every variable that appears as a positive literal in $t_n$, and $0$ to all other variables. We claim that $(\varphi, k) \in$ SHORTEST IMPLICANT CORE if and only if $(\mathcal{M}_{\varphi'}, \boldsymbol{x}, k) \in k$-SUFFICIENTREASON. For the forward direction, if there is an implicant $C \subseteq t_n$ of $\varphi$, of size at most $k$, then we claim that $C$ is also an implicant of each $\varphi^{(i)}$. This follows from the fact that every assignment $\sigma$ that is consistent with $C$ and satisfies $\varphi$, has a related assignment $\sigma^i$, that for every variable $x_j \in X_c$ assigns $\sigma^i(x_j^i) = \sigma(x_j)$, and that is equal to $\sigma$ for every $x_j \notin X_c$. It is clear that $\sigma^i(\varphi^{(i)}) = \sigma(\varphi)$, which concludes the claim. As $C$ is an implicant of each $\varphi^{(i)}$, it must also be an implicant of $\varphi'$. Then, as $\mathcal{M}_{\varphi'}$ is equivalent to $\varphi'$ (as Boolean functions) by construction, and $\boldsymbol{x}$ is consistent with $C$ because it is consistent with $t_n$, it follows that the partial instance that is induced by $C$ is a sufficient reason for $\boldsymbol{x}$ under $\mathcal{M}_{\varphi'}$. For the backward direction, assume there is a sufficient reason $\boldsymbol{y}$ for $\boldsymbol{x}$ under $\mathcal{M}_{\varphi'}$, whose size is at most $k$, and let $C'$ be its associated implicant for $\varphi'$. We cannot say yet that $C'$ is a proper candidate for being an implicant core of $\varphi$, as $C'$ could contain variables not mentioned by $t_n$. Let us define $X_c'$ to be the set of variables of $\varphi'$ that are not present in $t_n$. Intuitively, as there are $k + 1$ copies of each variable of $X_c'$ in $\varphi'$, no valuation of a variable in $X_c'$, for the formula $\varphi$, can be forced by a sufficient reason of size at most $k$. We will prove this idea in the following claim, allowing us to build an implicant $C$ for which we can assure $C \subseteq t_n$.

**Claim 22.** *Assume that there is an implicant $C'$ of size at most $k$ for $\varphi'$, and let $C$ be the partial valuation that sets every variable $x$ that appear in $t_n$ and that is defined by $C'$ to $C'(x)$, and that leaves every other variable undefined. Then $C'$ is an implicant of size at most $k$ for $\varphi$.*

*Proof.* The set $X_c'$ can be expressed as the union of $k + 1$ disjoint sets of variables, namely $X_c^1, \ldots, X_c^{k+1}$, where $X_c^i$ contains all variables of the form $x_j^i$. Since $C'$ contains at most $k$ literals, and there are $k+1$ disjoint sets $X_c^i$, there must exist an index $l$ such that $X_c^l \cap C' = \varnothing$. But then this implies that $C$ is an implicant of $\varphi^{(l)}$. But $\varphi^{(l)}$ is equivalent to $\varphi$ up to renaming of the variables that are not present in $C$, therefore, the fact that $C$ is an implicant of $\varphi^{(l)}$ implies that $C$ must be an implicant of $\varphi$ as well. $\qquad\square$

By using Claim 22 we get that $C$ is an implicant of $\varphi$. But we have that $C \subseteq t_n$, which is enough to conclude that $(\varphi, k) \in$ SHORTEST IMPLICANT CORE and finishes the proof of Lemma 21. $\qquad\square$

## Appendix D. Proof of Proposition 7

We now prove Proposition 7, whose statement we recall here:

**Proposition 7.** *The query* CHECKSUFFICIENTREASON *is (1) in* PTIME *for FBDDs, (2) in* PTIME *for perceptrons, and (3)* co-NP-*complete for MLPs.*

We prove each claim separately.

**Lemma 23.** *The query* CHECKSUFFICIENTREASON *can be solved in linear time for FBDDs.*

*Proof.* Let $(\mathcal{M}, \boldsymbol{x}, \boldsymbol{y})$ be an instance of the problem, with $\mathcal{M}$ being an FBDD. We first check that $\boldsymbol{x}$ is a completion of $\boldsymbol{y}$, which can clearly be done in linear time. We the define $\mathcal{M}'$ as the resulting FBDD from the following procedure: (i) For every internal node in $\mathcal{M}$ with label $i$, delete its outgoing edge with label 0 if $\boldsymbol{y}_i = 1$ and its outgoing edge with label 1 if $\boldsymbol{y}_i = 0$. We note here that $\mathcal{M}'$ is not a well defined FBDDs, since some internal nodes may have only one outgoing edge: more precisely, the value $\mathcal{M}(\boldsymbol{x}') \in \{0, 1\}$ is well defined for every instance $\boldsymbol{x}'$ that is a completion of $\boldsymbol{y}$, and is not defined for an instance $\boldsymbol{x}'$ that is not a completion of $\boldsymbol{y}$. To check whether $\boldsymbol{y}$ is a sufficient reason, we can then simply check that every leaf that is reachable from the root in $\mathcal{M}'$ is labeled the same (either true or false). This can clearly be done in linear time by standard graph algorithms. $\quad\square$

**Lemma 24.** *The query* CHECKSUFFICIENTREASON *can be solved in linear time for perceptrons.*

*Proof.* Let $(\mathcal{M} = (\boldsymbol{w}, b), \boldsymbol{x}, \boldsymbol{y})$ be an instance of the problem. We first check in linear time that $\boldsymbol{x}$ is a completion of $\boldsymbol{y}$. We then get rid of the components that are defined by $\boldsymbol{y}$, as follows. We define:

- $A := \sum_{y_i \neq \perp} y_i w_i$;
- $\boldsymbol{w}' := (w_i \mid y_i = \perp)$; and
- $b' := b + A$;

and let $\mathcal{M}'$ be the perceptron $(\boldsymbol{w}', b')$. Notice that the dimension of $\mathcal{M}'$ is equal to the number of undefined components of $\boldsymbol{y}$; we denote this number by $m$. It is then clear that $\boldsymbol{y}$ is a sufficient reason of $\boldsymbol{x}$ under $\mathcal{M}$ if and only if every instance of $\mathcal{M}'$ is labeled the same. We can check this as follows. Let $J_1$ be the minimum possible value of $\langle \boldsymbol{w}', \boldsymbol{x}' \rangle$ (for $\boldsymbol{x}' \in \{0, 1\}^m$); $J_1$ can clearly be computed in linear time by setting $x'_i = 0$ if $w'_i \geq 0$ and $x'_i = 1$ otherwise. Similarly we can compute the maximal possible value $J_2$ of $\langle \boldsymbol{w}', \boldsymbol{x}' \rangle$. Then, every instance of $\mathcal{M}'$ is labeled the same if and only if it is not the case that $J_1 < -b'$ and $J_2 \geq -b'$, thus concluding the proof. $\quad\square$

**Lemma 25.** *The query* CHECKSUFFICIENTREASON *is* co-NP-*complete for MLPs.*

*Proof.* We first show membership in co-NP. Let $(\mathcal{M}, \boldsymbol{x}, \boldsymbol{y})$ be an instance of the problem. Then $\boldsymbol{y}$ is a sufficient reason of $\boldsymbol{x}$ under $\mathcal{M}$ if and only if all the completions of $\boldsymbol{y}$ are labeled the same as $\boldsymbol{x}$. This can clearly be checked in co-NP.

In order to prove hardness we reduce from TAUT, the problem of checking whether an arbitrary boolean formula is a satisfied by all possible assignments of its variables. This problem is known to be complete for co-NP. Let $\mathcal{F}$ be an arbitrary boolean formula. We use Lemma 13 to build an equivalent MLP $\mathcal{M}$ in polynomial time (with the features of $\mathcal{M}$ corresponding to the variables of $\mathcal{F}$). Then $\mathcal{F}$ is a tautology if and only if all completions of the partial instance $\boldsymbol{y} = \perp^n$ are positive instances of $\mathcal{M}$. First, we construct an arbitrary instance $\boldsymbol{x}$ (for instance, the one with all the features being 0), and we reject if $\mathcal{M}(\boldsymbol{x}) = 0$. Then, we accept if $\boldsymbol{y}$ is a sufficient reason of $\boldsymbol{x}$ under $\mathcal{M}$, and we reject otherwise. This concludes the reduction. $\quad\square$

## Appendix E. Proof of Proposition 8

We prove Proposition 8, whose statement we recall here:

**Proposition 8.** *The query* COUNTCOMPLETIONS *is (1) in* PTIME *for FBDDs, (2)* #P-*complete for perceptrons, and (3)* #P-*complete for MLPs.*

As we said in the main text, the first claim follows almost directly from the definition of FBDDs; see [35] for instance. For the second claim, we will rely on the #P-hardness of the counting problem #Knapsack, as defined next:

**Definition 26.** *An input of the problem #Knapsack consists of natural numbers $s_1, \ldots, s_n, k \in \mathbb{N}$ (given in binary). The output is the number of subsets $S \subseteq \{1, \ldots, n\}$ such that $\sum_{i \in S} s_i \leq k$.*

The problem #Knapsack is well known to be #P-complete. Since we were not able to find a proper reference for this fact, we prove it here by using the #P-hardness of the problem #SubsetSum. An input of the problem #SubsetSum consists of natural numbers $s_1, \ldots, s_n, k \in \mathbb{N}$, and the output is the number of subsets $S \subseteq \{1, \ldots, n\}$ such that $\sum_{i \in S} s_i = k$. The problem #SubsetSum is shown to be #P-complete in [4, Theorem 4]. From this we can deduce:

**Lemma 27** (Folklore). *The problem #Knapsack is #P-complete.*

*Proof.* Membership in #P is trivial. We prove hardness by polynomial-time reduction from #SubsetSum. Let $(s_1, \ldots, s_n, k) \in \mathbb{N}^{n+1}$ be an input to #SubsetSum. It is clear that #SubsetSum$(s_1, \ldots, s_n, 0)$ = #Knapsack$(s_1, \ldots, s_n, 0)$, and that for $k \geq 1$ we have #SubsetSum$(s_1, \ldots, s_n, k)$ = #Knapsack$(s_1, \ldots, s_n, k)$ − #Knapsack$(s_1, \ldots, s_n, k-1)$, thus establishing the reduction. □

We can now show the second claim of Proposition 8.

**Lemma 28.** *The query* COUNTCOMPLETIONS *is #P-complete for perceptrons.*

*Proof.* Membership in #P is trivial. We show hardness by polynomial-time reduction from #Knapsack. Let $(s_1, \ldots, s_n, k)$ be an input of #Knapsack. Let $\mathcal{M}$ be the perceptron with weights $s_1, \ldots, s_n$ and bias $-(k+1)$. Remember that we consider only perceptrons that use the step activation function, so that an instance $\boldsymbol{x} \in \{0, 1\}^n$ is positive for $\mathcal{M}$ if and only if $\sum_{i=1}^{n} x_i s_i - (k+1) \geq 0$. It is then clear that #Knapsack$(s_1, \ldots, s_n, k)$ = $2^n$ − COUNTPOSITIVECOMPLETIONS$(\mathcal{M}, \perp^n)$, thus establishing the reduction. □

Finally, the third claim of Proposition 8 simply comes from the fact that MLPs can simulate arbitrary Boolean formulas (Lemma 13), and the fact that counting the number of satisfying assignments of a Boolean formula (#SAT) is #P-complete.

## Appendix F. Proof of Proposition 9

We now prove Proposition 9, that is:

**Proposition 9.** *The query* COUNTCOMPLETIONS *can be solved in pseudo-polynomial time for perceptrons (assuming the weights and biases to be integers given in unary).*

The first part of the proof is to show how to transform in polynomial time and arbitrary instance of COUNTPOSITIVECOMPLETIONS for perceptrons (with the weights and bias being integers given in unary) into an instance of #Knapsack that has the same number of solutions.

**Lemma 29.** *Let $\mathcal{M} = (\boldsymbol{w}, b)$ be a perceptron having at least one positive instance, with the weights and bias being integers given in unary, and let $\boldsymbol{x}$ be a partial instance. We can build in polynomial time an input $(s_1, \ldots, s_m, k) \in \mathbb{N}^{m+1}$ of #Knapsack such that* COUNTPOSITIVECOMPLETIONS$(\mathcal{M}, \boldsymbol{x})$ = #Knapsack$(s_1, \ldots, s_m, k)$, *with $s_1, \ldots, s_m, k$ written in unary (i.e., their value is polynomial in the input size).*

*Proof.* The first step is to get rid of the components that are defined by $\boldsymbol{x}$, like we did in Lemma 24. Define

- $A := \sum_{x_i \neq \perp} x_i w_i$;

- $\boldsymbol{w}' := (w_i \mid x_i = \perp)$; and

- $b' := b + A$;

and let $\mathcal{M}'$ be the perceptron $(\boldsymbol{w}', b')$. Notice that the dimension of $\mathcal{M}'$ is equal to the number of undefined components of $\boldsymbol{x}$; let us write $m$ this number. It is then clear that COUNTPOSITIVECOMPLETIONS$(\mathcal{M}, \boldsymbol{x})$ is equal to the number of positive instances of $\mathcal{M}'$, that is, of instances $\boldsymbol{x}' \in \{0, 1\}^m$ that satisfy

$$\langle \boldsymbol{w}', \boldsymbol{x}' \rangle + b' \geq 0 \tag{4}$$

Now, let $J$ be the maximum possible value of $\langle \boldsymbol{w}', \boldsymbol{x}' \rangle$; $J$ can clearly be computed in linear time by setting $x_i' = 1$ if $w_i' \geq 0$ and $\boldsymbol{x}_i' = 0$ otherwise. We then claim that the number of solutions to Equation 4 is equal to the number of solutions of

$$\langle \boldsymbol{s}, \boldsymbol{x}' \rangle \leq k, \tag{5}$$

where $s_i := |w_i'|$ for $1 \leq i \leq m$ and $k := J + b'$. Indeed, consider the function $h : \{0, 1\}^m \to \{0, 1\}^m$ defined componentwise by $h(x_i') := x_i'$ if $w_i' < 0$ and $h(x_i') := 1 - x_i'$ otherwise. Then $h$ is a bijection, and we will show that for any $\boldsymbol{x}' \in \{0, 1\}^m$, we have that $\boldsymbol{x}'$ satisfies Equation 4 if and only if $h(\boldsymbol{x}')$ satisfies Equation 5, from which our claim follows. In order to see this, consider that

$$(3) \iff \sum_i w_i' x_i' \geq -b' \iff \sum_{w_i \geq 0} w_i' x_i' + \sum_{w_i < 0} w_i' x_i' \geq -b' \tag{6}$$

$$\iff \sum_{w_i \geq 0} |w_i'| x_i' - \sum_{w_i < 0} |w_i'| x_i' \geq -b' \tag{7}$$

$$\iff \sum_{w_i < 0} |w_i'| x_i' - \sum_{w_i \geq 0} |w_i'| x_i' \leq b' \tag{8}$$

$$\tag{9}$$

On the other hand, we have

$$h(\boldsymbol{x}') \text{ satisfies } (4) \iff \sum_i |w_i'| h(x_i') \leq J + b' \tag{10}$$

$$\iff \sum_{w_i < 0} |w_i'| x_i' + \sum_{w_i \geq 0} |w_i'| (1 - x_i') \leq \sum_{w_i \geq 0} |w_i'| + b' \tag{11}$$

$$\iff (7) \tag{12}$$

Last, let us observe that we have $k \geq 0$, as otherwise $\mathcal{M}$ would not have any positive instance. Therefore $(s_1, \ldots, s_m, k)$ is a valid input of #Knapsack, which concludes the proof. $\square$

We can now easily combine Lemma 29 together with a well-known dynamic programming algorithm solving #Knaspsack in pseudo-polynomial time.

*Proof of Proposition 9.* Let $\mathcal{M} = (\boldsymbol{w}, b)$ be a perceptron, with the weights and bias being integers given in unary, and let $\boldsymbol{x}$ be a partial instance. First, we check that the maximal value of $\langle \boldsymbol{x}, \boldsymbol{w} \rangle$ is greater than $-b$, as otherwise $\mathcal{M}$ has no positive instance and we can simply return 0. We then use Lemma 29 to build in polynomial time an instance $(s_1, \ldots, s_m, k) \in \mathbb{N}^{m+1}$ of #Knapsack such that COUNTPOSITIVECOMPLETIONS$(\mathcal{M}, \boldsymbol{x}) = $#Knapsack$(s_1, \ldots, s_m, k)$, and with $s_1, \ldots, s_m, k$ being written in unary (i.e., their value is polynomial in the input size). We can then compute #Knapsack$(s_1, \ldots, s_m, k)$ by dynamic programming as follows. For $i \in \{1, \ldots, m\}$ and $C \in \mathbb{N}$, define the quantity $\mathrm{DP}[i][C] := |\{S \subseteq \{1, .., i\} | \sum_{j \in S} s_j \leq C\}|$. We wish to compute $\mathrm{DP}[m][k]$. We can do so by computing $\mathrm{DP}[i][C]$ for $i \in \{1, \ldots, m\}$ and $C \in \{0, \ldots, k\}$, using the relation $\mathrm{DP}[i + 1][C] = \mathrm{DP}[i][C] + \mathrm{DP}[i][C - s_{i+1}]$, and starting with the convention that $\mathrm{DP}[0][a] = 0$ for all $a < 0$ and that $\mathrm{DP}[0][a] = 1$ for all $a \geq 0$. It is clear that the whole procedure can be done in polynomial time. $\square$

## Appendix G. Proof of Proposition 10

We prove in this section Proposition 10, whose statement we recall here:

**Proposition 10.** *The problem* COUNTCOMPLETIONS *restricted to perceptrons admits an FPRAS (and the use of randomness is not even needed in this case). This is not the case for MLPs, on the other hand, at least under standard complexity assumptions.*

The fact that the query has no FPRAS for MLPs is because MLPs can efficiently simulate Boolean formulas (Lemma 13), and it is well known that the problem #SAT (of counting the number of satisfying assignments of a Boolean formula) has no FPRAS unless $NP = RP$. Hence we only need to prove our claim concerning perceptrons.

*Proof of Proposition 10 for perceptrons.* We can assume without loss of generality that the weights and bias are integers, as we can simply multiply every rational by the lowest common denominator (note that the bit lenght of the lowest common denominator is polynomial, and that it can be computed in polynomial time[5]). We then transform the perceptron and partial instance to an input of #Knapsack with the right number of solutions using Lemma 29, by observing that the construction also takes polynomial time when the input weights are given in binary (and by considering that the $s_1, \ldots, s_m, k$ are also computed in binary). We can then apply an FPTAS to this #Knapsack instance, as shown in [18, 29]. $\qquad\square$

## Appendix H.  Background in parameterized complexity

In this section we present the notions from parameterized complexity that we will need to prove Proposition 11.

A *parameterized problem* is a language $L \subseteq \Sigma^* \times \mathbb{N}$, where $\Sigma$ is a finite alphabet. For each element $(x, k)$ of a parameterized problem, the second component is called the *parameter* of the problem. A parameterized problem is said to be *fixed parameter tractable* (FPT) if the question of whether $(x, k)$ belongs to $L$ can be decided in time $f(k) \cdot |x|^{O(1)}$, where $f$ is a computable function.

The FPT class, as well as the other classes we will introduce in this paper, are closed under a particular kind of reductions. A mapping $\phi : \Sigma^* \times \mathbb{N} \to \Sigma^* \times \mathbb{N}$ between instances of a parameterized problem $A$ to instances of a parameterized problem $B$ is said to be an *fpt-reduction* if and only if

- $(x, k)$ is a yes-instance of $A$ $\iff$ $\phi(x, k)$ is a yes-instance of $B$.
- $\phi(x, k)$ can be computed in time $|x|^{O(1)} \cdot f(k)$;
- There exists a computable function $g$ such that $k' \leq g(k)$, where $k'$ is the parameter of $\phi(x, k)$.

We define the complexity classes that are relevant for this article in terms of circuits. Recall that a circuit is a rooted directed acyclic graph where nodes of in-degree 0 are called *input gates*, and that the root of the circuit is called the *output gate*. Internal gates can be either OR, AND, or NOT gates. All NOT nodes have in-degree 1. Nodes of types AND and OR can either have in-degree at most 2, in which case they are said to be *small* gates, or in-degree bigger than 2, in which case they are said to be *large* gates. The *depth* of a circuit is defined as the length (number of edges) of the longest path from any input node to the output node. The *weft* of a circuit is defined as the maximum amount of large gates in any path from an input node to the output node. An *assignment* of a circuit $C$ is a function from the set of input gates in $C$ to $\{0, 1\}$. The weight of an assignment is defined as the number of input gates that are assigned 1. Assignments of a circuit naturally induce a value for each gate of the circuit, computed according to the label of the gate. We say an assignment *satisfies* a circuit if the value of the output gate is 1 under that assignment.

The main classes we deal with are those composing the W-hierarchy and the W(Maj)- hierarchy, a variant proposed by Fellows et al. [14]. These complexity classes can be defined upon the WEIGHTED CIRCUIT SATISFIABILITY problem, parameterized by specific classes $\mathcal{C}$ of circuits, as defined below.

| | |
|---|---|
| Problem: | WEIGHTED CIRCUIT SATISFIABILITY($\mathcal{C}$), abbreviated WCS($\mathcal{C}$) |
| Input: | A circuit $C \in \mathcal{C}$ |
| Parameter: | An integer $k$ |
| Output: | YES, if there is a satisfying assignment of weight exactly $k$ for $C$, and NO otherwise. |

We consider two restricted classes of circuits. First, $C_{t,d}$, the class of circuits using the connectives AND, OR, NOT that have weft at most $t$ and depth at most $d$. On the other hand, we consider $M_{t,d}$, the class of circuits that use (only) the MAJORITY connective (that is satisfied exactly when more than half of its inputs are true), have weft at most $t$ and depth at most $d$. In the case of majority gates, we allow multiple parallel edges. Observe that, even though his is not useful for circuits with (OR, AND, NOT)-gates, it allows circuits majority gates to receive multiple times a same input. In the case of majority gates, a gate is said to be small if its fan-in is at most 3.

We can then define each class W[$t$] (resp., W(Maj)[$t$]) as the set of parameterized problems that can be fpt-reduced to WCS($C_{t,d}$) (resp., WCS($M_{t,d}$)) for some constant $d$. Note that the notion of *can be fpt-reduced* is transitive, and thus the classes W[$t$] and W(Maj)[$t$] are closed under fpt-reductions. As usual, a parameterized problem $A$ is then said to be W[$t$]-hard (resp., W(Maj)[$t$]-hard) when every parameterized problem in W[$t$] (resp., W(Maj)[$t$]) can be fpt-reduced to $A$.

## Appendix I. Proof of Proposition 11

In this section we prove Proposition 11, that is:

**Proposition 11.** *For every $t \geq 1$ the* MINIMUMCHANGEREQUIRED *query over rMLPs with $3t + 3$ layers is* W(Maj)[$t$]-*hard and is contained in* W(Maj)[$3t + 7$].

We first explain what are rMLPs, then sketch the proof, and then proceed with the proof.

Given an MLP $\mathcal{M}$, with the dimension of the layers being $d_0, \dots, d_k$, we define its *graph size* as $N := \sum_{i=0}^{k} d_i$. We say an MLP with graph size $N$ is restricted (abbreviated as rMLP) if each of its weights and biases can be represented as a decimal number with at most $O(\log(N))$ digits. More precisely, represented as $\sum_{i=-K}^{K} a_i 10^i$, for integers $0 \leq a_i \leq 9$ and $K \in O(\log N)$. Note that all numbers expressible in this way are also expressible by fractions, where the numerator is an arbitrary integer bounded by a polynomial in $N$, and the denominator is a power of 10 whose value is bounded as well by a polynomial in $N$.

We now explicit a family of parameterized problems indexed by an integer $t \geq 1$.

| | |
|---|---|
| Problem: | $t$-MINIMUMCHANGEREQUIRED, abbreviated $t$-MCR |
| Input: | An rMLP $\mathcal{M}$ with at most $t$ layers, an instance $\boldsymbol{x}$ |
| Parameter: | An integer $k$ |
| Output: | YES, if there exists an instance $\boldsymbol{y}$ with $\mathrm{d}(\boldsymbol{x}, \boldsymbol{y}) \leq k$ and $\mathcal{M}(\boldsymbol{x}) \neq \mathcal{M}(\boldsymbol{y})$, and NO otherwise |

We rewrite the statement of Proposition 11 with this explicit notation.

**(Restatement of Proposition 11).** *For every $t \geq 1$, the $(3t + 3)$-MCR problem is* W(Maj)[$t$]-*hard and is contained in* W(Maj)[$3t + 7$].

As the proof of Proposition 11 is quite involved, we first present a proof sketch that summarizes the process.

**Hardness.** We prove hardness in Section I.1. Showing that a parameterized problem $A$ is W[$t$]-hard (resp., W(Maj)[$t$]-hard) is usually complicated since, by directly using the definition, one would have to show that for every fixed $d \in \mathbb{N}$, there exists an fpt-reduction $f_d$ from WCS($C_{t,d}$) (resp., from WCS($M_{t,d}$)) to $A$. Instead, it is usually more convenient to prove first some form of *normalization theorem* stating that a particular class of circuits, for which one knows the value of $d$, is already hard for W[$t$] (or W(Maj)[$t$]).[6] Following this approach, we start by showing

loose normalization theorem for the $W(\mathsf{Maj})$-hierarchy in Lemma 30; namely, we prove that the problem $\mathrm{WCS}(M_{3t+2,3t+3})$ is $W(\mathsf{Maj})[t]$-hard. The main difficulty here is to reduce the depth $d$ of the majority circuits, for any fixed $d \in \mathbb{N}$, to a depth of at most $3t + 3$. We then show in Lemma 31 that rMLPs can simulate majority circuits, without increasing the depth of the circuit. In Theorem 32 we use this construction to show an fpt-reduction from $\mathrm{WCS}(M_{3t+2,3t+3})$ to $(3t + 3)$-MCR. This is enough to conclude hardness for $W(\mathsf{Maj})[t]$.

**Membership.** We prove membership in Section I.2. Presented in Theorem 34, the proof consists of 4 steps. We first show in Lemma 35 how to transform a given rMLP $\mathcal{M}$ that into an MLP $\mathcal{M}'$ that uses only step activation functions and that has the same number of layers. Then, as a second step, we build an MLP $\mathcal{M}''$, with $3t + 4$ layers and again using only the step activation function, such that $\mathcal{M}''$ has a satisfying assignment of weight $k$ if and only if $(\mathcal{M}, \boldsymbol{x}, k)$ is a positive instance of the $t$-MCR problem. The third step is to use a result of circuit complexity [17] stating that circuits with weighted thresholds gates (which are equivalent to biased step functions), can be transformed into circuits using only majority gates, increasing the depth by no more than 1. This yields a circuit $C_{\mathcal{M}''}$ with $3t + 5$ layers. However, the circuit $C_{\mathcal{M}''}$, resulting from the construction of Goldmann et al. [17], has both positive variables and negated variables as inputs, as their model needs to be able to represent non-monotone functions. For the fourth and last step, we build a circuit $C^*_{\mathcal{M}''}$ based on $C_{\mathcal{M}''}$, that fits the description of majority circuits as defined by [14, 15] (i.e., the one that we use). This circuit $C^*_{\mathcal{M}''}$ has weft $3t + 7$, and we prove that $(C^*_{\mathcal{M}''}, k + 1)$ is a positive instance of the Weighted Circuit Satisfiability problem that characterizes the class $W(\mathsf{Maj})[t]$ if and only if $(\mathcal{M}, \boldsymbol{x}, k)$ is a positive instance of the $(3t + 3)$-MCR problem. The whole construction being an fpt-reduction, this will be enough to conclude membership in $W(\mathsf{MAJ})[3t + 7]$.

Observe that (r)MLPs can be interpreted as well as rooted directed acyclic graphs, with weighted edges and where each node is associated a layer according to its (unweighted) distance from the root. Every node in a certain layer $\ell$ is connected to every node in layers $\ell - 1$ and $\ell + 1$. We will sometimes use this equivalent interpretation, which turns out to be more handy for some of the proofs in this section.

## I.1 Hardness

As explained in the proof sketch, we start by establishing a normalization theorem for the $W(\mathsf{Maj})$-hierarchy.

**Lemma 30.** *The problem* $\mathrm{WCS}(M_{3t+2,3t+3})$ *is* $W(\mathsf{Maj})[t]$-*hard.*

*Proof.* A significant part of this proof is based on techniques due to Fellows et al. [14] and to Buss et al. [6]. Let $C$ be an arbitrary majority circuit of weft at most $t$ and depth at most $d \geq t$ for some constant $d$, and let $k$ be the parameter of the input instance. We define a *small sub-circuit* as a maximally connected sub-circuit comprising only small gates. Now, consider a path $\pi$ from an arbitrary input node of $C$ to its output gate. We claim that $\pi$ intersects at most $t + 1$ small sub-circuits. Indeed, there must be at least one large gate separating every pair of small sub-circuits intersected by $\pi$, as otherwise the maximality assumption would be broken. But in $\pi$, as in any path, there are at most $t$ large gates, because of the weft restriction, from where we conclude the claim. Now, for each small sub-circuit $S$, consider the set $I_S$ of its inputs (that may be either large gates or input nodes of $C$). As small gates have fan-in at most 3, and the depth of each small sub-circuit is at most $d$, we have that $|I_S| \leq 3^d$. We can thus enumerate in constant time all the satisfying assignments of $S$. We identify each assignment with the set of variables to which it assigns the value 1. We keep a set $\Gamma$ with the satisfying assignments among $I_S$ that are minimal with respect to $\subseteq$. Then, because of the fact that majority circuits are monotone, $S$ can be written in monotone DNF as

$$S \equiv \bigvee_{\gamma \in \Gamma} \bigwedge_{x \in \gamma} x$$

Note that the size of $\Gamma$ is trivially bounded by the constant $2^{3^d}$. We then build a circuit $C'$, based on $C$, by following these steps:

1. Add $3^d(k + 1)$ extra input nodes. We distinguish the first, that we denote as $u$, from the $3^d(k + 1) - 1$ remaining, that we refer to by $N$.

2. Add a new output gate that is a binary majority between the old output gate and the node $u$.

3. Replace every small sub-circuit $S$ by its equivalent monotone DNF formula, consisting of one large OR-gate and many large AND-gates.

4. Relabel every large OR-gate, of fan-in $\ell \leq 2^{3^d}$ created in the previous step to be a majority gate with the same inputs, but to which one wires as well $\ell$ parallel edges from the input node $u$.

5. Relabel every large AND-gate $g$, of fan-in $\ell \leq 3^d$, to be a majority gate. If $g$ had edges from gates $g_1, \ldots, g_\ell$, then replace each edge coming from a $g_i$ by $k+1$ parallel edges, and finally, wire $\ell(k+1) - 1$ nodes in $N$ to $g$.

An illustration of the transformation ins presented in Figure 3. We now check that $C'$ is a (majority) circuit in $M_{3t+2,3t+3}$. To bound the depth and weft of $C'$ we need to account for all the sub-circuits of depth 2 that we introduced in steps 3–5 to replace each small sub-circuit of $C$. Note that two small sub-circuits that were parallel in $C$ (meaning no input-output path could intersect both) have corresponding sub-circuits that are parallel in $C'$. Consider now an arbitrary path $\pi$ from a variable to the root of $C$, and let $\pi'$ be the corresponding path in $C'$ (that goes to the new root of $C'$). The path $\pi$ contains one variable gate, at most $t$ large gates, and intersects at most $t+1$ small sub-circuits. The corresponding path $\pi'$ in $C'$ still contains the variable gate, the (at most $t$) large gates that were in $\pi$, and for each of the at most $t+1$ small-subcircuits that $\pi$ intersected, $\pi'$ now contains exactly 2 large gate (and $\pi'$ also contains the new output gate of $C'$). Therefore, the length of $\pi'$ is at most $1 + t + 2(t+1) + 1 - 1 = 3t + 3$, and it contains at most $t + 2(t+1) = 3t + 2$ large gates. Since every path $\pi'$ in $C'$ from a variable to the root of $C'$ corresponds to such a path $\pi$ in $C$, we obtain that the depth of $C'$ is at most $3t + 3$ and its weft is at most $3t + 2$. Hence, $C'$ is indeed a majority circuit in $M_{3t+2,3t+3}$.

We now prove that ($\star$) there is a satisfying assignment of weight $k + 1$ for $C'$ if and only if there is a satisfying assignment of weight $k$ for $C$, which would conclude our fpt-reduction. The proof for this claim is based on how the constructions in step 4 and 5 actually simulate large OR-gates and AND-gates, respectively.[7] We prove each direction in turn.

**Forward direction.** Let us assume that there exists a satisfying assignment of weight $k + 1$ for $C'$. First, because input node $u$ is directly connected to the output gate through a binary majority, it must be assigned to 1 in order to satisfy $C'$. Let $C''$ be the sub-circuit of $C'$ formed by all the nodes that descend from the old output-gate in $C'$. Then $C''$ needs to be satisfied in order to satisfy $C'$. Since $u$ is not present in $C''$, an assignment of weight $k + 1$ that satisfies $C'$ is made by assigning 1 to $u$ and to exactly $k$ other input gates. In order to prove the claim, we will show that ($\dagger$) an assignment of weight $k$ for the inputs of $C''$ satisfies $C''$ if and only if its restriction to the inputs of $C$ satisfies $C$, assuming $u$ is assigned to 1. As $C''$ only differs from $C$ because of the replacement of each small sub-circuit $S$ by its equivalent DNF, and the additional inputs in $N$, we only need to prove that steps 4 and 5 actually compute large OR and AND gates. Consider a gate $g$ introduced in step 4, having edges from gates $g_1, \ldots, g_\ell$ and $\ell$ edges from node $u$. Therefore, $g$ has fan-in $2\ell$, and as $u$ always contributes with a value of $\ell$ to $g$, we have that $g$ is satisfied exactly when at least one of the gates $g_1, \ldots, g_\ell$ is satisfied. Consider now a gate $g$ introduced in step 5. By construction, $g$ has fan-in equal to $2\ell(k+1) - 1$, from which we deduce that if all gates $g_1, \ldots, g_\ell$ are satisfied, then $g$ is indeed satisfied in $C''$. On the other hand, if an assignment of weight $k$ does not satisfy every gate $g_i$, then $g$ receives at most $(\ell - 1)(k + 1)$ units from the gates $g_i$, and as the assignment has weight $k$, it receives at most $k$ from the nodes in $N$. Thus, $g$ receives at most $(k + 1)\ell - 1$ units, which is less than half of its fan-in, and thus, $g$ is not satisfied. Thus, we have proved ($\dagger$). However, notice that the restriction of the assignment might have a weight of strictly less than $k$ in $C$. But it is clear that, since the circuit is monotone, we can increase the weight by setting some variables of $C$ to 1, until the weight becomes equal to $k$. This proves the forward direction.

**Backward direction.** Let us now assume an assignment of weight $k$ for $C$. We then we extend such an assignment to $C'$ by assigning 0 to the inputs in $N$ and 1 to $u$. Thanks to ($\dagger$), this is a

(a) A majority circuit where *small sub-circuits* are represented with blue blobs, and black nodes correspond to large majority gates. The path determining the *weft* is colored red. The longest path, determining the *depth* of the circuit, is drawn with a dashed orange line.

(b) The majority circuit where *small sub-circuits* have been replaced by depth-2 majority circuits, corresponding to their equivalent DNF. The equivalent DNF depth-2 sub-circuits are represented by rectangles. Once again, the path determining the *weft* is colored red. The longest path, determining the *depth* of the circuit, is drawn with a dashed orange line.

Figure 3: Illustration of the Normalization Lemma (30). In a nutshell, by paying a controlled increase in weft, the depth of the circuit can be substantially reduced.

satisfying assignment of weight $k + 1$ for $C'$, which proves the backward direction of $(\star)$ and thus concludes the proof of Lemma 30. □

Then, we show that rMLPs can simulate majority circuits, without increasing the depth of the circuit.

**Lemma 31.** *Given a circuit $C$ containing only majority gates, we can build in polynomial time an rMLP that is equivalent to $C$ (as a Boolean function) and whose number of layers is equal to the depth of $C$.*

*Proof.* First, note that we can assume that circuit $C$ does not contain parallel edges by replacing each gate $g$ having $p$ edges to a gate $g'$ by $p$ copies $g_1, \ldots, g_p$ with single edges to $g'$. We then build a layerized circuit (remember the definition of a layerized circuit from Appendix A) $C'$ from $C$, by applying the same construction that we used in Lemma 13 to layerize a circuit, but using unary majority gates as identity gates instead. Note that the depth of $C'$ is the same as that of $C$.

Next, we show how each non-output majority gate can be simulated by using two relu-gates (again, remember the definition of a relu gate from Appendix A). First, note that (†) for any non-negative integers $x, n \in \mathbb{N}$, the function

$$f_n(x) := \text{relu}\left(x - \lfloor \frac{n}{2} \rfloor\right) - \text{relu}\left(x - \lfloor \frac{n}{2} \rfloor - 1\right)$$

is equal to

$$\text{Maj}_n(x) = \begin{cases} 1 & \text{if } x > \frac{n}{2} \\ 0 & \text{otherwise} \end{cases}.$$

We will use (†) to transform the majority circuit $C'$ into a circuit $C''$ that has only relu gates for the non-output gates, and that is equivalent to $C'$ in a sense that we will explain next. For every non-output majority gate $g$ of $C'$, we create two relu gates $g_1', g_2'$ of $C''$. The idea is that (⋆) for any valuation of the input gates (we identify the input gates of $C'$ with those of $C''$), the Boolean value of any non-output gate $g$ in $C'$ will be equal to the (not necessarily Boolean) value of gate $g_1'$ (in $C''$) minus the value of the gate $g_1'$ (in $C''$). We now explain what the biases of these new gates $g_1', g_2'$ for every majority gate $g$ of $C'$ are. Letting $n$ be the in-degree of a majority gate $g$ in $C'$, the bias of $g_1'$ is $-\lfloor \frac{n}{2} \rfloor$, and that of $g_2'$ is $-\lfloor \frac{n}{2} \rfloor - 1$. Next, we explain what the weights of these new gates $g_1', g_2'$ are and how we connect them to the other relu gates. We do this by a bottom-up induction on $C'$, that is, on the level of the gates of $C'$ (since $C'$ is layerized), and we will at the same time show that (⋆) is satisfied. To connect the gates $g_1', g_2'$ to the gates of the preceding layer, we differentiate two cases:

**Base case.** The inputs of the gate $g$ are variable gates; in other words, the level of $g$ in $C'$ is 1 (remember that variable gates have level 0). We then set these variable gates to be an input of both $g_2'$ and $g_2'$, and set all the weights to 1. It is clear that (⋆) is satisfied for the gates $g, g_1', g_2'$, thanks to (†).

**Inductive case.** The inputs of the gate $g$ are other majority gates; in other words, the level of $g$ in $C'$ is $> 1$. Then, let $^1g, \ldots, {}^m g$ be the inputs[8] (majority gates) of the gate $g$ in $C'$, and consider their associated pairs of relu gates $(^1g_1', {}^1 g_2'), \ldots, (^m g_1', {}^m g_2')$ in $C''$. We then set all the gates $^1g_1', \ldots, {}^m g_1$ to be input gates of both gates $g_1'$ and $g_2'$, with a weight of 1, and set all the gates $^1g_2, \ldots, {}^m g_2$ to be input gates of both gates $g_1'$ and $g_2'$, with a weight of $-1$. By induction hypothesis, and using again (†), it is clear that (⋆) is satisfied.

Finally, based on the output gate $r$ of $C'$, we create a step gate $r'$ in $C''$ in the following way. Let $^1g, \ldots, {}^m g$ be the inputs of $r$, and $(^1g_1', {}^1 g_2'), \ldots, (^m g_1', {}^m g_2')$ their associated pairs in $C''$. Then wire each gate $^i g_1'$ to $r'$ with weight 1, and also wire each gate $^i g_2'$ to $r'$ with weight $-1$. Let $-\lfloor \frac{n}{2} \rfloor - 1$ be the bias of $r'$.

We have constructed a circuit $C''$ whose output gate is a step gate, and all other gates are relu gates. Consider now a valuation $\boldsymbol{x}$ of the input gates of $C'$, which we identify as well as a valuation $\boldsymbol{x}'$ of the input gates of $C''$. We claim that $C'(\boldsymbol{x}) = 1$ if and only if $C''(\boldsymbol{x}') = 1$. But this simply comes from the fact that for $x, n \in \mathbb{N}$, we have $x > \frac{n}{2} \iff x \geq \lfloor \frac{n}{2} \rfloor + 1$, and from the fact that (⋆) is satisfied for the input gates of $r$ and of $r'$.

The last thing that we have to do is to transform the circuit $C''$, that uses only relu gates except for its output step gate, into a valid MLP. This can be done easily as in the proof of Lemma 13 by adding dummy connections with weights zero, because $C''$ is layerized. The resulting MLP $\mathcal{M}_C$ is then equivalent to $C$, it is clearly an rMLP, its number of layers is exactly the depth of $C$, and, since we have constructed it in polynomial time, this concludes the proof. □

Finally, we use this construction to show an fpt-reduction from $\mathrm{WCS}(M_{3t+2,3t+3})$ to $(3t+3)$-MCR. This is enough to conclude hardness for $\mathrm{W}(\mathsf{Maj})[t]$, thanks to Lemma 30.

**Theorem 32.** *There is an fpt-reduction from the problem* $\mathrm{WCS}(M_{3t+2,3t+3})$ *to the* $(3t+3)$-*MCR problem.*

*Proof.* We will in fact show an fpt-reduction from $\mathrm{WCS}(M_{t,t})$ to $t$-MCR, which gives the claim when applied to $3t + 3$, noting of course that $\mathrm{WCS}(M_{3t+3,3t+3})$ is trivially at least as hard as $\mathrm{WCS}(M_{3t+2,3t+3})$. Let $(C, k)$ be an instance of $\mathrm{WCS}(M_{t,t})$. We first build an MLP $\mathcal{M}_C$ equivalent to $C$ (as Boolean functions) by using Lemma 31. The MLP $\mathcal{M}_C$ has $t$ layers. Then, we build an MLP $\mathcal{M}_C'$, that is based on $\mathcal{M}_C$, by following the steps described below:

1. Initialize $\mathcal{M}_C'$ to be an exact copy of $\mathcal{M}_C$.

2. Add an extra input, that we call $v_1$, to $\mathcal{M}_C'$. This means that if $\mathcal{M}_C$ had dimension $n$, then $\mathcal{M}_C'$ has dimension $n + 1$.

3. Create nodes $v_2, \ldots, v_t$, all having a bias of 0, and for each $1 \le i < t$, connect node $v_i$ to node $v_{i+1}$ with an edge of weight 1.

4. Let $r$ be the root of $\mathcal{M}'_C$, and let $m$ be its fan-in. We connect node $v_t$ to $r$ with an edge of weight $m$. Moreover, if the bias of $r$ in $\mathcal{M}_C$ was $b$, we set it to be $b - m$ in $\mathcal{M}'_C$.

5. Observe that $\mathcal{M}'_C$ is layerized. To make it a valid MLP (where all the neurons of a layer are connected to all the neurons of the adjacent layers), we do as in the proof of Lemma 13 by adding dummy null weights.

It is clear that the construction of $\mathcal{M}'_C$ takes polynomial time, and that its number of layers is again $t$. We now prove a claim describing the behavior of $\mathcal{M}'_C$.

**Claim 33.** *For any instance $\boldsymbol{x}'$ of $\mathcal{M}'_C$, expressed as the concatenation of a feature $\boldsymbol{x}'_1$ (for the extra input node $v_1$) and an instance $\boldsymbol{x}$ of $\mathcal{M}_C$, we have that $\boldsymbol{x}'$ is a positive instance of $\mathcal{M}'_C$ if and only if $\boldsymbol{x}'_1 = 1$ and $\boldsymbol{x}$ is a positive instance of $\mathcal{M}_C$*

*Proof.* Consider that, by construction, an instance $\boldsymbol{x}'$ is positive for $\mathcal{M}'_C$ if and only if

$$\sum_{i=1}^{n+1} \boldsymbol{h}'^{(t-1)}_i \boldsymbol{W}'^{(t)}_i = m\boldsymbol{h}'^{(t-1)}_1 + \sum_{i=2}^{n+1} \boldsymbol{h}'^{(t-1)}_i \boldsymbol{W}'^{(t)}_i \ge -b + m$$

But by construction $\boldsymbol{h}'^{(t-1)}_1 = \boldsymbol{x}'_1$, and $\sum_{i=2}^{m+1} \boldsymbol{h}'^{(t-1)}_i \boldsymbol{W}'^{(t)}_i = \sum_{i=1}^{m} \boldsymbol{h}^{(t-1)}_i \boldsymbol{W}^{(t)}_i$. This means that $\boldsymbol{x}'$ is a positive instance of $\mathcal{M}'_C$ if and only if

$$m\boldsymbol{x}'_1 + \sum_{i=1}^{m} \boldsymbol{h}^{(t-1)}_i \boldsymbol{W}^{(t)}_i \ge -b + m$$

Note that if $\boldsymbol{x}'_1 = 1$ and $\boldsymbol{x}$ is a positive instance of $\mathcal{M}_C$, this inequality is achieved, making $\boldsymbol{x}'$ a positive instance. For the other direction, it is clear that it holds if $\boldsymbol{x}'_1 = 1$. We show that in fact $\boldsymbol{x}'_1 = 0$ is not possible. Indeed, by the construction of $\mathcal{M}_C$, we have that $0 \le \sum_{i=1}^{m} \boldsymbol{h}^{(t-1)}_i \boldsymbol{W}^{(t)}_i \le m$, and also that $-b \ge 1$, which makes the inequality unfeasible.

This concludes the proof of the claim. $\qquad\square$

This claim has two important consequences:

1. As satisfying assignments of $C$ correspond to positive instance of $\mathcal{M}_C$, we have that there is a satisfying assignment of weight exactly $k$ for $C$ if and only if there is a positive instance of weight exactly $k + 1$ for $\mathcal{M}'_C$.

2. The instance $0^{n+1}$ is negative for $\mathcal{M}'_C$

This consequences will allow us to finish the reduction. Consider the instance $(\mathcal{M}'_C, 0^{n+1}, k + 1)$ of $t$-MCR. We claim that this is a positive instance for the problem if and only if $(C, k)$ is a positive instance of $\text{WCS}(M_t)$.

For the forward direction, consider $(\mathcal{M}'_C, 0^{n+1}, k + 1)$ to be a positive instance of $t$-MCR. This means there is an instance $\boldsymbol{x}^*$ that has the opposite classification as $0^{n+1}$ under $\mathcal{M}'_C$, and differs from it in at most $k + 1$ features. By the second consequence of the claim, $\boldsymbol{x}^*$ must be a positive instance. Also, differing in at most $k + 1$ features from $0^{n+1}$ means that $\boldsymbol{x}^*$ has weight at most $k + 1$. But as majority gates are monotone connectives, majority circuits are monotones as well, so the existence of a positive instance $\boldsymbol{x}^*$ of weight at most $k + 1$ implies the existence of a positive instance $\boldsymbol{x}'^*$ of weight exactly $k + 1$. Therefore, by the first consequence of the claim, there is a satisfying assignment of weight exactly $k$ for $C$, which implies $(C, k)$ is a positive instance of $\text{WCS}(M_{t,t})$

For the backward direction, consider $(C, k)$ to be a positive instance of $\text{WCS}(M_{t,t})$. This means, by the first consequence of the claim, that there is a positive instance $\boldsymbol{x}^*$ of weight exactly $k + 1$ for $\mathcal{M}'_C$. But based on the second consequence of the claim, $0^{n+1}$ is a negative instance for $\mathcal{M}'_C$.

As $\boldsymbol{x}^*$ differs from $0^{n+1}$ in no more than $k+1$ features, and they have opposite classifications, we have that $(\mathcal{M}'_C, 0^{n+1}, k+1)$ is a positive instance of $t$-MCR.

As the whole construction takes polynomial time, and the reduction changes the parameter in a computable way, from $k$ to $k+1$, it is an fpt-reduction. This concludes the proof. $\qquad\square$

## I.2 Membership

In this section we prove membership in $\mathrm{W}(\mathsf{Maj})[3t+7]$. This will be enough to prove:

**Theorem 34.** *There is an fpt-reduction from $t$-MCR to $WCS(M_{t+4,t+4})$, implying $(3t+3)$-MCR belongs to $\mathrm{W}(\mathsf{Maj})[3t+7]$.*

As explained in the proof sketch, we first show how to transform a given rMLP $\mathcal{M}$ that into an MLP $\mathcal{M}'$ that uses only step activation functions and that has the same number of layers. More formally, we prove that rMLPs using only step activation functions are powerful enough to simulate MLPs that use relu activation functions in the internal layers (and a step function for the output neuron). The construction is polynomial in the width (maximal number of neurons in a layer) of the given relu-rMLP, but exponential on its depth (number of layers). We show:

**Lemma 35.** *Given an rMLP $\mathcal{M}$ with relu activation functions, there is an equivalent MLP $\mathcal{M}'$ that uses only step activation functions and has the same number of layers. Moreover, if the number of layers of $\mathcal{M}$ is bounded by a constant, then $\mathcal{M}'$ can be computed in polynomial time.*

*Proof.* Let $(\boldsymbol{W}^{(1)}, \ldots, \boldsymbol{W}^{(\ell)})$, $(\boldsymbol{b}^{(1)}, \ldots, \boldsymbol{b}^{(\ell)})$ and $(f^{(1)}, \ldots, f^{(\ell)})$ be the sequences of weights, biases, and activation functions of the rMLP $\mathcal{M}$. Note that $f^{(i)}$ for $1 \le i \le \ell-1$ is relu and that $f^{(\ell)}$ is the step activation function. The first step of the proof is to transform every weight and bias into an integer. To this end, let $L \in \mathbb{N}$, $L > 0$ be the lowest common denominator of all the weights and biases, and let $\mathcal{M}'$ be the MLP that is exactly equal to $\mathcal{M}$ except that all the weights have been multiplied by $L$, and all the biases of layer $i$ have been multiplied by $L^i$. Observe that $\mathcal{M}'$ has only integer weights and biases. When $w$ (resp., $b$) is a weight (resp., bias) of $\mathcal{M}$, we write $w'$ (resp., $b'$) the corresponding value in $\mathcal{M}'$. We claim that $\mathcal{M}$ and $\mathcal{M}'$ are equivalent, in the sense that for every $\boldsymbol{x} \in \{0,1\}^n$, it holds that $\mathcal{M}(\boldsymbol{x}) = \mathcal{M}'(\boldsymbol{x})$. Indeed, for $0 \le i \le \ell$, let $\boldsymbol{h}^{(i)}$ and $\boldsymbol{h}'^{(i)}$ be the vectors of values for the layers of $\mathcal{M}$ and $\mathcal{M}'$, respectively, as defined by Equation 1. We will show that $(\star)$ for all $1 \le i \le \ell-1$ we have $\boldsymbol{h}'^{(i)} = L^i \times \boldsymbol{h}^{(i)}$. The base case of $i=0$ (i.e., the inputs) is trivially true. For the inductive case, assume that $(\star)$ holds up to $i$ and let us show that it holds for $i+1$. We have:

$$
\begin{aligned}
\boldsymbol{h}'^{(i+1)} &= \mathrm{relu}(\boldsymbol{h}'^{(i)}\boldsymbol{W}'^{(i+1)} + \boldsymbol{b}'^{(i+1)}) \\
&= \mathrm{relu}(L \times \boldsymbol{h}'^{(i)}\boldsymbol{W}^{(i+1)} + L^{i+1} \times \boldsymbol{b}^{(i+1)}) \text{ by the definition of } \mathcal{M}' \\
&= \mathrm{relu}(L^{i+1} \times \boldsymbol{h}^{(i)}\boldsymbol{W}^{(i+1)} + L^{i+1} \times \boldsymbol{b}^{(i+1)}) \text{ by inductive hypothesis} \\
&= L^{i+1} \times \mathrm{relu}(\boldsymbol{h}^{(i)}\boldsymbol{W}^{(i+1)} + \boldsymbol{b}^{(i+1)}) \text{ by the linearity of relu} \\
&= L^{i+1} \times \boldsymbol{h}^{(i+1)},
\end{aligned}
$$

and $(\star)$ is proven. Since the step function (used for the output neuron) satisfies $\mathrm{step}(cx) = c\,\mathrm{step}(x)$ for $c > 0$, we indeed have that $\mathcal{M}(\boldsymbol{x}) = \mathcal{M}'(\boldsymbol{x})$.

We now show how to build a model $\mathcal{M}''$ that uses only step activation functions and that is equivalent to $\mathcal{M}'$. The first step is to prove an upper bound for the values in $\boldsymbol{h}'$. We start by bounding the values in $\boldsymbol{h}$. Let $D$ be width of $\mathcal{M}$, that is, the maximal dimension of a layer of $\mathcal{M}$, and let $C$ be the maximal absolute value of a weight or bias in $\mathcal{M}$; note that the value of $C$ is asymptotically bounded by $|\mathcal{M}|^{O(1)}$ because $\mathcal{M}$ is an rMLP. For every instance $\boldsymbol{x}$, we have that

$$
0 \le h_j^{(i)} = \mathrm{relu}\left(\sum_k h_k^{(i-1)}W_{k,j}^{(i)} + b_j^{(i)}\right) \le DC \max_k h_k^{(i-1)} + C \le (D+1)C \max(1, \max_k h_k^{(i-1)})
$$

Using this inequality, and the fact that $\max_k h_k^{(0)} \le 1$, we obtain inductively that $0 \le h_j^{(i)} \le ((D+1)C)^i$. By $(\star)$, this implies that $0 \le h_j'^{(i)} \le ((D+1)CL)^i$.

Figure 4: Illustration of the conversion from a relu activation function to step activation functions, for $S = 3$. The weights are unchanged, and if the bias of the original neuron was $b$ then the bias in the $j$-th copy of that neuron becomes $b - j$.

As all values (weights, biases and the $\boldsymbol{h}'$ vectors) in $\mathcal{M}'$ consist only of integers, and are all bounded by the integer $S := ((D+1)CL)^{\ell}$, then each relu in $\mathcal{M}'$ with bias $b$ becomes equivalent to the following function $f^*$:

$$f^*(x + b) := [x + b \geq 1] + [x + b \geq 2] + \ldots + [x + b \geq S] \tag{13}$$

Where $[y \geq j] := 1$ if $y \geq j$ and 0 otherwise. Hence, in order to finish the proof, it is enough to show how activation functions of the form $f^*$ can be simulated with step activation functions. Namely, we show how to build $\mathcal{M}''$, that uses only step activation functions, from $\mathcal{M}'$, in such a way that both models are equivalent. In order to do so, we replace each $f^{(i)}, \boldsymbol{W}'^{(i)}, \boldsymbol{b}'^{(i)}$ for $1 \leq i \leq \ell$ in the following way. If $i = \ell$, then nothing needs to be done, as $f^{(\ell)}$ is already assumed to be a step activation function. When $1 \leq i < \ell$, we replace the weights, activations and biases in a way that is better described in terms of the underlying graph of the MLP. We split every internal node, with bias $b$ into $S$ copies, all of which will have the same incoming and outgoing edges as the original nodes, with the same weights. The $j$-th copy will have a bias equal to $b - j$. We illustrated this step in Figure 4. This construction is an exact simulation of the function $f^*$ defined in Equation 13.

The computationally expensive part of the algorithm is the replacement of each node in $\mathcal{M}'$ by $S$ nodes, which takes time at most $S = ((D+1)CL)^{\ell} \in O(|\mathcal{M}|^{\ell}(CL)^{\ell})$ per node and thus at most $O(|\mathcal{M}|^{\ell+1}(CL)^{\ell})$ in total. Since $\ell$ is a constant, and $C$ is bounded by a polynomial on $\mathcal{M}$, we only need to argue that $L$ is bounded as well. Indeed, as $\mathcal{M}$ is an rMLP, each weight and bias can be assumed to be represented as a fraction whose denominator is a power of 10 of value polynomial in the graph size $N$ of $\mathcal{M}$. But the lowest common multiple of a set of powers of 10 is exactly the largest power of 10 in the set. Therefore $L \leq 10^p$, where $p \in O(\log N)$, and thus $L \in O(N^c) \subseteq O(|\mathcal{M}|^c)$ for some constant $c$. We conclude from this that the construction takes polynomial time. $\quad\square$

We are now ready to prove Theorem 34.

*Proof of Theorem 34.* Let $(\mathcal{M}, \boldsymbol{x}, k)$ be an instance of $t$-MCR. During this reduction we assume that $n > 2k$, as otherwise the result can be achieved trivially; if $n \leq 2k$ then trying all instances that differ by at most $k$ from $\boldsymbol{x}$ takes only $O(k^k)$, and thus we can solve the entire problem in fpt-time and return a constant-size instance of $WCS(M_{t+2})$, completing the reduction.

We start by applying Lemma 35 to build an equivalent MLP $\mathcal{M}'$ that uses only step activation functions. As $t$ is constant, this construction takes polynomial time, and its resulting MLP $\mathcal{M}'$ has $t$ layers as well. If $\boldsymbol{x}$ is a negative instance of $\mathcal{M}'$ (and thus of $\mathcal{M}$) we do nothing. This can trivially be checked in polynomial time, evaluating $\boldsymbol{x}$ in $\mathcal{M}'$. But if $\boldsymbol{x}$ happens to be a positive instance of $\mathcal{M}'$,

then we change the definition of $\mathcal{M}'$ negating its root perceptron[9], and thus making $\boldsymbol{x}$ a negative instance. As a result, we can safely assume $\boldsymbol{x}$ to be a negative instance of $\mathcal{M}'$. We can also, in the same fashion that we assumed $n > 2k$, discard the case where the instance $0^n$ is a positive instance of $\mathcal{M}'$ that differs by at most $k$ from $\boldsymbol{x}$, as in such scenario we could also solve the problem in fpt-time. The same can be done for $1^n$.

We now build an MLP $\mathcal{M}''$, that still uses only step activation functions, such that $\mathcal{M}''$ has a positive instance of weight exactly $k$ if and only if $(\mathcal{M}, \boldsymbol{x}, k)$ is a positive instance of $t$-MCR.

Let $\mathcal{M}''$ be a copy of $\mathcal{M}'$ to which we add one extra layer at the bottom. For each $1 \leq i \leq n$, we connect the $i$-th input node of $\mathcal{M}''$ to what was the $i$-th input node of $\mathcal{M}'$, but is now an internal node in $\mathcal{M}''$. If $\boldsymbol{x}_i = 0$ then the node in $\mathcal{M}''$ corresponding to the $i$-th input node of $\mathcal{M}'$ has a bias of 1, and the weight of the edge coming from the $i$-th input node of $\mathcal{M}''$ is also 1. On the other hand, if $\boldsymbol{x}_i = 1$, then the node in $\mathcal{M}''$ corresponding to the $i$-th input node of $\mathcal{M}'$ has a bias of 0, and the weight of the connection added to it is $-1$. After doing this, we add $k - 1$ more input nodes to $\mathcal{M}''$, a new node $p$ in the $t$-th layer and a new root node $r''$, that is placed in the layer $t + 1$. We connect $r'$, the previous root node, to $r''$ of $\mathcal{M}'$ with weight 1, and all input nodes to node $p$ with weights of 1. In case $p$ is more than one layer above the new input nodes, we connect them through paths of identity gates, as shown in Lemma 13. We set the bias of $r''$ to $-2$, and the bias of $p$ to $-k$. All non-input nodes added in the construction use step activation functions.

We now prove a claim stating that $\mathcal{M}''$ has exactly the intended behavior.

**Claim 36.** *The MLP $\mathcal{M}''$ has a positive instance of weight exactly $k$ if and only if $(\mathcal{M}, \boldsymbol{x}, k)$ is a positive instance of $t$-MCR.*

*Proof.* For the forward direction, assume $\mathcal{M}''$ has a positive instance $\boldsymbol{x}'$ of weight exactly $k$. As the root $r''$ has a bias of $-2$, and two incoming edges with weight 1, and given that the output of any node is bounded by 1, as only step activation functions are used, we conclude that both $p$ and $r'$, the children of $r''$, must have a value of 1 on $\boldsymbol{x}'$. The fact that $r'$ has a value of 1 on $\boldsymbol{x}'$ implies that $\boldsymbol{x}^s$, the restriction of $\boldsymbol{x}$ that considers only nodes that descend from $r'$, must be a positive instance for the submodel $\mathcal{M}^s$ induced by considering only nodes that descend from $r'$. But one can easily check that by construction, we have that $\mathcal{M}^s(\boldsymbol{x}^s) = \mathcal{M}'(\boldsymbol{x}^s \oplus \boldsymbol{x})$, where $\oplus$ represents the bitwise-xor. Thus, $\boldsymbol{x}^s \oplus \boldsymbol{x}$ is a positive instance for $\mathcal{M}$, and consequently for $\mathcal{M}$. As $\boldsymbol{x}^s \oplus \boldsymbol{x}$ differs from $\boldsymbol{x}$ by exactly the weight of $\boldsymbol{x}^s$, as 0 is the neutral element of $\oplus$, and the weight of $\boldsymbol{x}^s$ is by definition no more than the weight of $\boldsymbol{x}'$, which is in turn no more than $k$ by hypothesis, we conclude that $(\mathcal{M}, \boldsymbol{x}, k)$ is a positive instance of $t$-MCR.

For the backward direction, assume there is a positive instance $\boldsymbol{x}'$ of $\mathcal{M}$ that differs from $\boldsymbol{x}$ in at most $k$ positions. This means that $\boldsymbol{x}'' = \boldsymbol{x} \oplus \boldsymbol{x}'$ has weight at most $k$. By the same argument used in the forward direction, $\mathcal{M}^s(\boldsymbol{x}'') = \mathcal{M}'(\boldsymbol{x}'' \oplus \boldsymbol{x}) = \mathcal{M}'(\boldsymbol{x}')$, as $\boldsymbol{x} \oplus \boldsymbol{x}' \oplus \boldsymbol{x} = \boldsymbol{x} \oplus \boldsymbol{x} \oplus \boldsymbol{x}' = \boldsymbol{x}'$, because $\oplus$ is both commutative and its own inverse. But the fact that $\boldsymbol{x}'$ is a positive instance of $\mathcal{M}$ implies that it is also a positive instance for $\mathcal{M}'$. As we are assuming $\boldsymbol{x}'| \neq 0^n$, we have that $k - |\boldsymbol{x}'| \leq k - 1$. Thus, we can create an instance $\boldsymbol{x}''$ for $\mathcal{M}''$ that is equal to $\boldsymbol{x}'$ on its corresponding features, and that sets $k - |\boldsymbol{x}'|$ arbitrary extra input nodes to 1, among those created in the construction of $\mathcal{M}''$. As the instance $\boldsymbol{x}''$ has weight exactly $k$, it satisfies the submodel descending from $p$, and as $\boldsymbol{x}''$ its equal to $\boldsymbol{x}'$ on the submodel descending from $r'$, and $\boldsymbol{x}'$ is a positive instance of $\mathcal{M}'$, we have that this submodel must be satisfied as well. Both submodels being satisfied, the whole model $\mathcal{M}''$ is satisfied, hence we conclude the proof. $\square$

We thus have a model $\mathcal{M}''$ with step activation functions, and $t + 2$ layers, such that if that model has a satisfying assignment of weight exactly $k$, then $(\mathcal{M}, \boldsymbol{x}, k)$ is a positive instance of $t$-MCR.

Note that step activation functions with bias are equivalent to weighted threshold gates. We then use a result by Goldmann and Karpinski [17, Corollary 12] to build a circuit $C_{\mathcal{M}''}$ that is equivalent (as Boolean functions) to $\mathcal{M}''$ but uses only majority gates. The construction of Goldmann et al. can be carried in polynomial time, and guarantees that $C_{\mathcal{M}''}$ will have at most $t + 3$ layers.

There is however a caveat to surpass: although not explicitly stated in the work of Goldmann et al. [17], their definition of majority circuit must assume that for representing a Boolean function from $\{0,1\}^n$ to $\{0,1\}$, the circuit is granted access to $2n$ input variables $x_1, \ldots, x_n, \overline{x_1}, \ldots, \overline{x_n}$, as it is usual in the field, and described for example in the work of Allender [1]. We thus assume that the circuit $C_{\mathcal{M}''}$ resulting from the construction of Goldmann et al. has this structure, which does not match the required structure of the majority circuits defining the W(Maj)-hierarchy as defined by Fellows et al [14, 15]. In order to solve this, we adapt a technique from Fellows et al. [15, p. 17]. We build a circuit $C_{\mathcal{M}''}^*$ that does fit the required structure. Let $n$ be the dimension of $\mathcal{M}''$ (which exceeds by $k-1$ that of $\mathcal{M}$). We now describe the steps one needs to apply to $C_{\mathcal{M}''}$ in order to obtain $C_{\mathcal{M}''}^*$.

1. Add a new layer with $n+1$ input nodes $x_1', \ldots, x_{n+1}'$, below what previously was the layer of $2n$ input nodes $x_1, \ldots, x_n, \overline{x_1}, \ldots, \overline{x_n}$.

2. For every $1 \leq i \leq n$, connect input node $x_i'$ with its corresponding node $x_i$ in the second layer, making $x_i$ a unary majority, with the same outgoing edges it had as an input node. This enforces $x_i = x_i'$.

3. Create a new root $r'$ for the circuit, and let $r'$ be a binary majority between the input node $x_{n+1}'$ and the previous root $r$.

4. Replace each previous input node $\overline{x_i}$ by a majority gates $m_i$ that has $n+1-2k$ incoming edges from $x_{n+1}'$, and one incoming edge from each $x_j'$ with $j \notin \{i, n+1\}$. The outgoing edges are preserved.

It is clear that the circuit $C_{\mathcal{M}''}^*$ is a valid majority circuit in the sense defining the W(Maj)-hierarchy. And it has 2 layers more than $C_{\mathcal{M}''}$, yielding a total of $t+5$ layers, where the last one has a small gate. However, it is not evident what this new circuit does. We now prove a tight relationship between the circuit $C_{\mathcal{M}''}^*$ and $\mathcal{M}''$.

**Claim 37.** *The circuit $C_{\mathcal{M}''}^*$ has a satisfying assignment of weight exactly $k+1$ if and only if $\mathcal{M}''$ has a positive instance of weight exactly $k$.*

*Proof.* **Forward Direction.** Assume $C_{\mathcal{M}''}^*$ has a satisfying assignment of weight $k+1$. By step 3 of the construction, in order to satisfy $C_{\mathcal{M}''}^*$, the assignment must set $x_{n+1}'$ to 1.

As we assume that node $x_{n+1}'$ is set to 1, the assignment must set to 1 exactly $k$ input nodes among $x_1', \ldots, x_n'$ and thus the sum of inputs set to 1 of each majority gate $m_i$ constructed in step 4, is exactly equal to

$$n + 1 - 2k + \sum_{j \notin \{i, n+1\}} x_j' = n + 1 - 2k + (k - x_i') = n + 1 - k - x_i'$$

and its fan-in is exactly equal to $2n - 2k$. Therefore $m_i$ is activated when $n + 1 - k - x_i' > n - k$, which happens precisely when $x_i' = 0$. This way, each gate $m_i$ corresponds to the negation of $x_i'$.

This way, the subcircuit induced by considering only the nodes that descend from $r'$ computes the same Boolean function that $C_{\mathcal{M}''}$ computes, under the natural mapping of their variables. Therefore, a satisfying assignment of weight $k+1$ for $C_{\mathcal{M}''}^*$ implies the existence of a satisfying assignment for $C_{\mathcal{M}''}$ that chooses exactly $k$ positive variables, and thus a positive instance of weight $k$ for $\mathcal{M}''$.

**Backward Direction.** Assume $\mathcal{M}''$ has a positive instance of weight exactly $k$. That implies that $C_{\mathcal{M}''}$ has a satisfying assignment $\sigma$ that sets at most $k$ positive variables to 1. Let us consider the assignment $\sigma'$ for $C_{\mathcal{M}''}^*$ that sets to 1 the same variables that $\sigma$ does, and additionally sets $x_{n+1}$ to 1. The assignment $\sigma'$ has weight exactly $k+1$. By the same argument used in the forward direction, under assignment $\sigma'$ the gates $m_i$ behave like negations. Thus, the assignment $\sigma'$ induces an assignment over the second layer of $C_{\mathcal{M}''}^*$ that corresponds precisely to a satisfying assignment of $C_{\mathcal{M}''}$, and thus makes the value of $r$ equal to 1. As both $r$ and $x_{n+1}$ have value 1 under assignment $\sigma'$, it follows that the value of $r'$, and thus of circuit $C_{\mathcal{M}''}^*$, are 1 under $\sigma'$ as well. This means that assignment $\sigma'$, which by construction has weight $k+1$, is a satisfying assignment for $C_{\mathcal{M}''}^*$, and thus concludes the proof. $\qquad\square$

By combining Claim 36 and Claim 37, and noting again that circuit $C^*_{\mathcal{M}''}$ is a valid majority circuit, in the sense that defines the $\mathrm{W}(\mathsf{Maj})$-hierarchy, and has weft at most $t+4$, we conclude the reduction of Theorem 34. $\qquad\square$

## Appendix J. Proof of Proposition 12

Based on Proposition 11, we know that interpreting an rMLP (for the problem MCR) with $9t + 27 = 3(3t+8)+3$ is $\mathrm{W}(\mathsf{Maj})[3t+8]$-hard. On the other hand, by using the same proposition, the problem of interpreting an rMLP with $3t + 3$ layers is contained in $\mathrm{W}(\mathsf{Maj})[3t + 7]$. But by hypothesis, $\mathrm{W}(\mathsf{Maj})[3t + 7] \subsetneq \mathrm{W}(\mathsf{Maj})[3t + 8]$, which is enough to conclude the proof.

## Footnotes

[3]We slightly abuse notation and write $x_u$ for the value of the feature of $\boldsymbol{x}$ that is indexed by the label of $u$.

[4]Note that, in order to keep our notation consistent, we use the symbol $\subseteq$ where Umans uses $\supseteq$.

[5]We need to compute the least common multiple (lcm) of a set of integers $a_1, \ldots, a_n$. Indeed, it is easy to check that $lcm(a_1, \ldots, a_n) = lcm(lcm(a_1, \ldots, a_{n-1}), a_n)$, which reduces inductively the problem to computing the lcm of two numbers in polynomial time. It is also easy to check that $lcm(a_1, a_2) = \frac{a_1 a_2}{gcd(a_1, a_2)}$, where $gcd(a_1, a_2)$ is the greatest common divisor of $a_1$ and $a_2$. As multiplication can clearly be carried in polynomial time, and Euclid's algorithm allows computing the $gcd$ function in polynomial time, we are done.

[6]Useful normalization theorems for the W-hierarchy are proved in the work of Downey, Fellows and Regan [11, 13], or Buss and Islam. [6]. Our normalization theorem for the W(Maj)-hierarchy is inspired from those.

[7]Although this technique can already be found in the work of Fellows et al. [14], we include it here for completeness.

[8] Please excuse us for using left superscripts.

[9]Let $\mathcal{P} = (\boldsymbol{w}, \boldsymbol{b})$ be the perceptron at the root of $\mathcal{M}'$, which contains only integer values by construction. Then, the negation of $\mathcal{P}$ is simply $\overline{\mathcal{P}} = (-\boldsymbol{w}, -\boldsymbol{b} + 1)$, as $-\boldsymbol{w}\boldsymbol{x} \geq -\boldsymbol{b} + 1$ precisely when $\boldsymbol{w}\boldsymbol{x} \leq \boldsymbol{b} - 1$, which occurs over the integers exactly when it is not true that $\boldsymbol{w}\boldsymbol{x} \geq \boldsymbol{b}$.