[Reviews · NeurIPS 2020]

Review 1

Summary and Contributions: The paper formalizes notions of explanations for FBDDs, perceptrons and ReLU networks in XAI as computational problems and studies the computational complexity of these problems in terms of standard complexity classes and parameterized complexity.

Strengths: From the point of view of computational complexity theory, interpretability is not much studied yet. Classifying the complexity of explanations is a natural step. Therefore this paper is a welcome contribution. The big picture that for perceptrons and decision diagrams problems are easy, but for general neural networks are hard, is well known. I think several results of the paper are folklore, but nevertheless it is useful to have those results carefully formulated and presented. The most interesting, novel and relevant results are the parameterized complexity results in Section 5. Parameterized complexity could be discussed in more detail, especially the W(Maj) hierarchy, which is not widely known and could have other uses in deep learning.

Weaknesses: An important concern with the paper is that the problems introduced in the paper are closely related to problems studied under different names in the literature, and there should be more discussion of these connections. This applies especially to ``sufficient reason''. A certificate of an input for a Boolean function is a well-known concept, which even has a section in the textbook of Arora-Barak cited by the paper (and is closely related to Huang's sensitivity theorem, a recent sensational result on Boolean functions). A certificate is the same as a sufficient reason in the paper, even the meanings of the terms are not that different. This kind of duplication is often an impediment for the development of research, especially for Boolean functions, which come up in many different contexts. Minimum change problems have also been studied before (though, as far as I know, not in the same form as here).

Correctness: Yes.

Clarity: The paper is very clearly written.

Relation to Prior Work: Please see above. NB: the author response promises to improve the presentation in this respect.

Reproducibility: Yes

Additional Feedback: The paper addressed the comments in a satisfactory manner.


Review 2

Summary and Contributions: The paper proposes to measure the interpretability of different machine learning models in terms of the computational complexity of solving certain problems associated with explaining the classification of an example. It presents four concrete problems (queries) and determines their complexity for 3 standard ML models: FBDDs, perceptrons, and multi-layer perceptrons. In the models which are intuitively less interpretable, the problems have higher computational complexity.

Strengths: This is a very well-written paper that takes a thoughtful approach to an important problem. The theoretical results are sound. The use of parametrized complexity to distinguish between different depth MLPs is nice.

Weaknesses: - A number of the proofs (given in the supplementary material) are straightforward. -The authors should make clearer which theoretical results required novel proofs and which are routine and/or follow easily from previous work. - It's not surprising that problems involving MLPs have higher complexity than problems involving FBDDs and perceptrons. - The paper of Darwiche and Marquise on the Knowledge Compilation Map should be discussed in related work. -The complexity of the minimum sufficient reason question does not seem like a particularly good measure for interpretability. It's not clear why computing a sufficient reason would need to be "minimum". Also, consider e.g. monotone DNF, which could be regarded as an easily interpretable representation. It's easy to to compute a minimum sufficient reason for a positive output, but NP-hard for a negative output.

Correctness: The results appear to be correct and carefully done (I did not check the details of all the proofs).

Clarity: Very well written.

Relation to Prior Work: Discussion of previous work is mostly good, but misses some relevant work in knowledge compilation. Darwiche and Marquis wrote a paper in 2002 entitled A Knowledge Compilation map that is closely related to this paper. It covers many representations of Boolean functions including FBDDs (not just OBDDs) and considers the complexity of a number of different queries including implication (same as check sufficient reason) and counting (closely related to count completions). In the paper, they state that implication and counting can both be done in poly time for FBDDs, and they provide justification for the statement with appropriate citations.

Reproducibility: Yes

Additional Feedback: Shortest implicant is Sigma_2^P complete for Boolean formulas. So it isn't necessary to reduce from Shortest Implicant Core, unless you want to prove something about depth 2 MLPs.


Review 3

Summary and Contributions: The paper sets out to find a characterization of the concept of interpretability of models, and propose one in terms of computational complexity by focusing on specific kinds of post-hoc queries.

Strengths: -- Important goal of pinpointing an elusive concept. -- Novel and sound complexity results. -- Relevance to the NeurIPS audience.

Weaknesses: -- The link between interpretability and complexity is made by definition, but not analyzed for adequacy. -- Related to the previous point, significance is questionable.

Correctness: All claims are sound to the extent that I checked.

Clarity: The manuscript is well written, and the authors do a good job of making the material reasonably self-contained.

Relation to Prior Work: In relation to my main concern (see below), the concept of interpretability does not seem to by linked in any way to complexity; a broader discussion of the literature might shed some light on this matter.

Reproducibility: Yes

Additional Feedback: As mentioned above, my main concern is with the basic idea of linking -- by definition -- model interepretability with computational complexity. There does not seem to be a well-founded basis for doing this, since interpretability is, to the best of my knowledge, an inherently subjective concept. This does not mean that studying the relationship between the two is useless, but perhaps it would be more interesting to set out from a more neutral position of looking for a correlation between them and seeing where the results lead. In its current form, the conclusions seem to be largely theoretical (as the authors themselves state); this is of course not inherently a bad thing, but it's very difficult to evaluate their significance. Minor comment: Footnote 1 may be avoided or made clearer by pointing out that their exists a decision problem complexity class that "corresponds" to #P, called PP, that allows to make a fair comparison. *** AFTER REBUTTAL I have revised my score to reflect a minor change of view after reading the authors' rebuttal and the comments by other reviewers.

[Author Response · NeurIPS 2020]

We thank the reviewers for their comments. All reviewers agree that our approach on exploring computational complexity as a way of comparing model interpretability would be of interest to the NeurIPS community, which is very encouraging for us. **Reviewer 1** and **Reviewer 2** also welcomed our formal approach to it (**R1**: *"Classifying the complexity of explanations is a natural step. Therefore this paper is a welcome contribution"*, **R2**: *"a thoughtful approach to an important problem"*). Both reviewers mention as a strong point the use of parameterized complexity when analyzing explanations for MLPs. **R1** even encourages us to present in more detail the classes that we use in our results as they *"could have other uses in deep learning"* (because of space limitations it was difficult to do so in the current version of the paper, but we would definitely add it in an extended version). We deal with specific comments by the reviewers below.

**Reviewer 4** appreciates that we take the risk of formally studying the elusive notion of interpretability (**R4**: *"Important goal of pinpointing an elusive concept"*), also praising our theoretical results (**R4**: *"Novel and sound complexity results"*). Nevertheless, the reviewer questions the significance of the approach, which can be summarized in the following comment: *"There does not seem to be a well-founded basis for [linking model interpretability with computational complexity], since interpretability is, to the best of my knowledge, an inherently subjective concept."* We agree with the reviewer: interpretability is largely a subjective concept and the community is far from having a definitive answer for what interpretability exactly means. But we actually consider this subjectivity as our main motivation: we wanted to explore ways into which some formal ground can be given to this subjective concept, even if in a preliminary form. This concern can thus be reduced to our choice of computational complexity as such a possible ground. **R4** considers this choice as *"not well-founded"* while the other two reviewers consider the study of the computational complexity of explanations as a plausible choice/step. We think that our results in the paper are a preliminary proof of this plausibility/adequacy, as the complexity of explanation queries actually correlates with the informal views on interpretability that can be found in the literature.

**R4** also questions our position in the following comment: *"[...] perhaps it would be more interesting to set out from a more neutral position of looking for a correlation between [complexity and interpretability]."* We disagree with this comment. Neutrality was a central concern when we were wording our submission and we were very careful in not claiming that computational complexity is the right or only way to understand interpretability. This is why we phrased our results explicitly as *"correlations"* between complexity and interpretability (exactly as the reviewer suggests). For instance, in the Abstract (line 4) we write that *"We make a step towards such a notion by studying whether folklore interpretability claims have a correlate in terms of computational complexity"*, and also in the Introduction with a similar comment (line 46). We were also careful in naming our notion as *complexity-based interpretability* when we formalize it (Section 2, line 110) to make it explicit that we are not formalizing a general notion of interpretability, but exploring a (new) narrow yet formal one. Although we disagree with the reviewer's view, this comment pushes us to make our position even more explicit in the paper. For the next version of the paper, we will include an additional discussion about this in the **Limitations** section and move this section to be right after the Introduction (it is the last section in the current submission).

We now focus on the comments by **R1** and **R2**. Both reviewers point out that some of the problems we treat, or slight variations of them, have already been discussed in the literature over different settings and under different names. We agree that our paper would improve by adding a discussion of the different names and contexts under which the literature has discussed similar problems, and explicitly mentioning their differences with our setting. We were under space constraints but in case of acceptance we will use the additional space to improve our Related Work section. **R1** and **R2** also suggest that our paper would benefit from being more explicit about the novelty and difficulty of some of the proofs by distinguishing those that involve new insights (e.g., parameterized complexity) from those that follow by already used techniques for the models that we consider. We will also implement this change in the next version. Thank you very much for the comments and the pointers. We appreciate **R1** pointing out certificate complexity and the study of boolean sensitivity as related subjects that could make for interesting lines of future research.

**R2** also comments about our choice of minimum sufficient reason (MSR) as one of the explainability queries as *"it's not clear why it would need to be minimum"*. We consider the minimality as a desirable property since an explanation (as a sufficient reason) can always be padded with superfluous information while remaining valid. Our rationale was that, among the different minimal sufficient reasons, it is arguably better to provide explanations that are as succinct as possible (observe that an input $x$ is always a sufficient reason for its own classification, but this is not really interesting). This comment also pushes us to provide a more technical discussion of the relevance of the chosen queries in the final version of our paper. Unfortunately due to space constraints we cannot address other interesting comments by **R2**, like the relationship between monotone DNFs and MSR, that suggests a further study of asymmetric models (for which MSR for positive and negative examples would have different complexities), or the need of using SHORTEST IMPLICANT CORE instead of just SHORTEST IMPLICANT. But we thank the reviewer as both comments give us the opportunity to improve our paper.

[Meta-Review · NeurIPS 2020]

Two referees strongly support the paper, one referee sees this paper marginally below the acceptance threshold. After re-reading the paper, the discussions and the rebuttal, I come to the conclusion that this paper does contain relevant contributions to the problem of formalizing the concept of interpretability by exploring connections with computational complexity. Further, I think that most points of criticism have been addressed in a convincing way in the rebuttal. So I recommend acceptance.